# Powerful and Theoretically Guaranteed Independence Testing on Heterogeneous Federated Clients

**Yixin Ren** [* 1]  **Hongquan Liu** [* 1]  **Juncai Zhang** [2 3]  **Yewei Xia** [1]  **Zichuan Lin** [4]  **Deheng Ye** [4]
**Hao Zhang** [2]  **Jihong Guan** [† 5]  **Shuigeng Zhou** [† 1]

## Abstract

In this paper, we present a novel federated independence testing (FedIT) method that addresses both theoretical and practical challenges arising from client heterogeneity. We begin by revisiting existing federated independence testing methods and showing why they fail to provide valid guarantees or maintain statistical power under data distributional shift across clients. Building on this analysis, we develop a copula-based marginal alignment technique together with a stacking-based aggregation strategy that amplifies intra-client dependence while mitigating inter-client variation, resulting in a theoretically sound and powerful global test. For practicality, we further accelerate the aggregation step and incorporate a privacy-preserving mechanism. On the theoretical side, we prove both the correctness of our method and the validity of the test. Empirically, we conduct extensive experiments on both synthetic and real-world datasets, which demonstrate the superiority of our solution over existing methods.

## 1. Introduction

Testing statistical independence is a foundational task in machine learning and statistics, supporting causal discovery (Hoyer et al., 2008; Zhang et al., 2012), representation learning (Li et al., 2021), and feature selection (Camps-Valls et al., 2010). Given observations sampled from a joint distribution $\mathbb{P}_{XY}$, the goal of *independence testing* (IT) is to determine whether $\mathbb{P}_{XY}$ factorizes as the product of its marginals, i.e., whether $\mathbb{P}_{XY} = \mathbb{P}_X \mathbb{P}_Y$ which is equivalent to $X$ and $Y$ being independent. As data volume expands and governance tightens, data are frequently siloed across institutions. For instance, hospitals keep their own patient records (Kidd et al., 2023), which cannot be pooled together due to privacy and security concerns, and regulation requirements. This situation motivates the *federated independence testing* (FedIT), which is to determine the independence relationship among variables without sharing raw data.

Compared with the centralized and well-studied *independent and identically distributed* (*i.i.d.*) setting for independence testing (Gretton et al., 2005; Zhang et al., 2012; Székely et al., 2007), FedIT is considerably more challenging because heterogeneity may degrade test power (Huang et al., 2020). To the best of our knowledge, there are only a few works that explicitly address this issue. The most recent is (Li et al., 2024), hereafter FUIT, which proposes a kernel-based federated test that accelerates computation via random features (Rahimi & Recht, 2007) and aggregates covariance-based summary statistics in the random feature space. Although FUIT achieves competitive results for federated causal discovery, we noticed from deep analysis that substantial headroom remains: its aggregation strategy is actually equivalent to naively concatenating samples in the feature space, thereby ignoring cross-client heterogeneity, lacking rigorous theoretical guarantees, and risking power deterioration under distribution shift. These limitations call for a theoretically grounded redesign of FedIT together with heterogeneity-aware aggregation mechanisms. Please see Appendix B for a more detailed review of related work.

This paper proposes a novel method for federated independence testing that tackles both theoretical and practical challenges posed by client heterogeneity. To ensure theoretical validity and practical robustness under heterogeneous marginal distributions and dependence structures, we introduce a unified framework that combines copula-based marginal alignment with a stacking-based aggregation mechanism. The copula-based alignment exploits an important property that copulas separates marginal distributions from dependence structures, as formalized by

---

[*]Equal contribution  [1]College of Computer Science snd Artificial Intelligence, Fudan University, Shanghai, China [2]Shenzhen Institutes of Advanced Technology, Chinese Academy of Sciences, Shenzhen, China [3]Department of Mathematics, Shantou University, Shantou, China [4]Tencent Hunyuan [5]School of Computer Science and Technology, Tongji University, Shanghai, China. Correspondence to: Jihong Guan <jhguan@tongji.edu.cn>, Shuigeng Zhou <sgzhou@fudan.edu.cn>.

*Proceedings of the 43^{rd} International Conference on Machine Learning*, Seoul, South Korea. PMLR 306, 2026. Copyright 2026 by the author(s).

Sklar's theorem (Sklar, 1959). This theorem ensures that any multivariate distribution can be uniquely decomposed into its marginals and a copula that captures their dependence (Nelsen, 2006). By mapping local data into a shared copula space, our method preserves the dependence structure while mitigating discrepancies in the marginals. Complementing this, the stacking-based aggregation strategy further enhances local dependence signals at each client and selectively integrates them based on their contributions to global test power. To ensure efficiency and privacy, we design a fast and privacy-preserving aggregation protocol, making the method effective in real federated settings.

Our main contributions are summarized as follows: 1) We provide a systematic analysis of the challenges in FedIT under client heterogeneity and propose a novel method that addresses both theoretical and practical challenges. 2) We introduce a copula-based marginal alignment technique combined with a stacking-based aggregation strategy that amplifies intra-client dependence while mitigating inter-client variation, resulting in a theoretically sound and powerful global test. To enhance its practicality, we further develop a fast and privacy-preserving aggregation protocol. 3) We provide theoretical guarantees on the correctness of our method and the validity of the test. 4) We empirically validate the proposed method on both synthetic and real-world benchmarks, demonstrating their practical effectiveness, and superiority over existing methods.

## 2. Preliminaries and Problem Formulation

We begin by recalling the classical hypothesis-testing framework for statistical independence, which serves as the foundation of federated independence testing (FedIT), and then formalize the federated setting with heterogeneous clients.

**Hypothesis testing framework.** The goal of independence testing is to decide whether two random variables $X$ and $Y$ are independent ($X \perp\!\!\!\perp Y$). Formally,

$$\mathcal{H}_0 : \mathbb{P}_{XY} = \mathbb{P}_X \mathbb{P}_Y \text{ versus } \mathcal{H}_1 : \mathbb{P}_{XY} \neq \mathbb{P}_X \mathbb{P}_Y. \quad (1)$$

The testing procedure is as follows: First, define the statistic $\rho$ and compute its empirical estimate from the samples. Next, choose a significance level $\alpha$ (typically set to 0.05). After that, obtain the corresponding $p$-value, which is the probability that the sampling of $\rho$ under $\mathcal{H}_0$ is at least as extreme as the observed value. Finally, the null hypothesis $\mathcal{H}_0$ is rejected if the $p$-value is not greater than $\alpha$.

Two types of errors may be generated during hypothesis testing. Type I error means the false rejection of $\mathcal{H}_0$, and Type II error indicates when $\mathcal{H}_0$ is wrong but not rejected. A good independence test (Gretton et al., 2007) requires that Type I error rate is upper bounded by $\alpha$ meanwhile maximizing the testing power ($1-$Type II error rate).

**Federated setting with heterogeneous clients.** We consider a federated setting with $K$ clients (distinct domains). Client $k \in [K]$ [1] holds $n_k$ samples $\boldsymbol{Z}_k := \{(x_i^{(k)}, y_i^{(k)})\}_{i=1}^{n_k}$ drawn from a joint distribution $\mathbb{P}_{X_k Y_k}$ on $X_k \in \mathbb{R}^{d_x}$ and $Y_k \in \mathbb{R}^{d_y}$. Let $\mathbb{P}_{X_k}$ and $\mathbb{P}_{Y_k}$ denote the corresponding marginals. All samples are mutually independent both intra- and inter-client. In FedIT and causal discovery (Huang et al., 2020; Li et al., 2024), it is common to assume that, although data distributions may vary by client, the dependence relationship between the variables is consistent. This reflects a mechanism-invariance view across clients.

**Assumption 2.1** (Consistent dependence assumption). The dependence relationship between $X_k$ and $Y_k$ is consistent across clients. Specifically, either $X_k \perp\!\!\!\perp Y_k$ holds for all $k \in [K]$, or $X_k \not\perp\!\!\!\perp Y_k$ holds for all $k \in [K]$.

This assumption reflects many real-world federated applications (e.g., multi-hospital studies, cross-region deployments), where a shared common data-generating mechanism governs all domains, even as local conditions differ without flipping the underlying dependence status.

**Assumption 2.2** (Heterogeneous clients assumption). The dependence mechanism (e.g., dependence strength or functional relationship), the marginal distributions $\mathbb{P}_{X_k}, \mathbb{P}_{Y_k}$ and the joint distributions $\mathbb{P}_{X_k, Y_k}$ may vary across clients.

Together, these assumptions define a realistic yet challenging regime: the global dependence status is common, but local distributions are heterogeneous. Our goal is to design a test that aggregates cross-client evidence to infer the shared dependence status while handling client heterogeneity.

## 3. Limitations of Existing FedIT Methods

In this section, we begin by identifying the aggregation challenges of federated independence testing (FedIT) under client heterogeneity and then show that existing methods not only face fundamental theoretical limitations but can also suffer substantial power loss in realistic scenarios.

### 3.1. Heterogeneous Aggregation Challenges

Fig. 1 summarizes two key challenges in FedIT. On the left, we illustrate the pitfall of naive aggregation strategies caused by heterogeneous marginal distributions. Two failure modes can occur: (i) independence relationships within individual clients lead to spurious dependence after aggregation; (ii) dependence relationships within individual clients lead to spurious independence after aggregation.

On the right, we highlight a practical challenge stemming from heterogeneous functional relationships across clients. Consider a simple example: Client 1 has variables $(X_1, Y_1)$

---

[1]Throughout, the symbol $[m]$ denotes the set $\{1, 2, ..., m\}$.

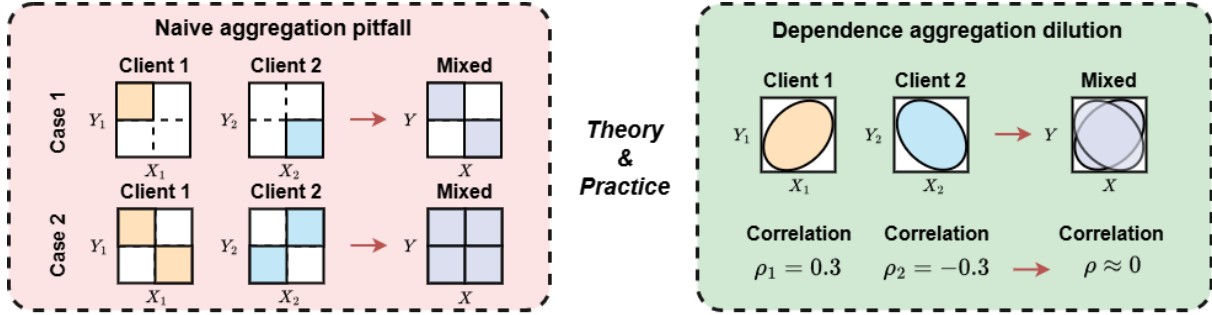

*Figure 1.* Overview of two key theoretical and practical aggregation challenges in heterogeneous FedIT. (Left) Naive aggregation pitfall: local independence may appear dependent after aggregation, while local dependence may cancel out and appear independent. (Right) Dependence dilution: strong local dependencies can be weakened after aggregation due to client heterogeneity.

following a bivariate Gaussian distribution with correlation coefficient 0.3, while Client 2 has variables $(X_2, Y_2)$ following a bivariate Gaussian distribution with correlation coefficient $-0.3$. After aggregation, these opposing correlations may cancel each other out, yielding an apparently uncorrelated global relationship. This scenario poses a major difficulty for independence testing, as it requires detecting higher-order dependencies beyond linear correlation, which in turn reduces the statistical power of the test.

Building on this, we revisit existing methods and show that they fail to address both the theoretical aggregation pitfall and the practical signal-dilution challenge described above.

### 3.2. Revisiting Existing FedIT Methods

We focus on a representative class of FedIT methods (Li et al., 2024) that extend the Kernel-based Independence Test (KIT) (Zhang et al., 2012) to the federated setting, referred to as FUIT. We show that FUIT's aggregation strategy is essentially equivalent to naively concatenating client samples in the feature space; consequently, it cannot overcome the above limitations under heterogeneity.

To see this, we first recall the KIT statistic in the centralized setting. Formally, let $\boldsymbol{x} = (\boldsymbol{x}_1, \boldsymbol{x}_2, \ldots, \boldsymbol{x}_n) \in \mathbb{R}^{d_x \times n}$ denote $n$ samples with dimension $d_x$, and define $\boldsymbol{y} \in \mathbb{R}^{d_y \times n}$ analogously. The KIT statistic is defined as $T = n\|C_{xy}\|_F^2$, where the covariance matrix is $C_{xy} := \frac{1}{n}\tilde{\phi}(\boldsymbol{x})^T \tilde{\phi}(\boldsymbol{y}) \in \mathbb{R}^{h \times h}$. Here, $\tilde{\phi}(\boldsymbol{x}) = \mathbf{H}\phi(\boldsymbol{x}) \in \mathbb{R}^{n \times h}$ is the centered random feature matrix, with $\mathbf{H} = \mathbf{I} - \frac{1}{n}\mathbf{1}\mathbf{1}^\top$ and $\mathbf{1} \in \mathbb{R}^{n \times 1}$ being the all-one vector. The random feature map is $\phi(\boldsymbol{x}) := \sqrt{2/h}\left[\cos(w_1^T \boldsymbol{x} + b_1); \ldots; \cos(w_h^T \boldsymbol{x} + b_h)\right]^T \in \mathbb{R}^{n \times h}$ where $w_i \sim \mathbb{P}(w)$ and $b_i \sim \text{Uniform}(0, 2\pi)$. Here, $h$ denotes the number of random features. The same construction applies to $\phi(\boldsymbol{y})$. With the statistic defined, the next step is to characterize its behavior under the null hypothesis.

To determine the rejection threshold, KIT approximates the null behavior of $T$ by fitting a two-parameter Gamma distribution under $\mathcal{H}_0$, whose shape and scale parameters

are determined by matching the first two moments:

$$
\begin{aligned}
\mathcal{E}_0 &:= \mathbb{E}_{\mathcal{H}_0}[T] = \text{Tr}(C_{xx}) \cdot \text{Tr}(C_{yy}), \\
\mathcal{V}_0 &:= \text{Var}_{\mathcal{H}_0}[T] = 2\|C_{xx}\|_F^2 \cdot \|C_{yy}\|_F^2,
\end{aligned}
\tag{2}
$$

where $C_{xx}$ and $C_{yy}$ are defined analogously as $C_{xy}$. Then the null distribution can be modeled as

$$
\mathcal{H}_0 : n\|C_{xy}\|_F^2 \sim \frac{x^{\gamma-1}e^{-x/\beta}}{\beta^\gamma \Gamma(\gamma)}, \quad \gamma = \frac{\mathcal{E}_0^2}{\mathcal{V}_0}, \ \beta = \frac{\mathcal{V}_0}{\mathcal{E}_0}, \tag{3}
$$

Hence, the critical threshold $\widehat{c_\alpha}$ is then can be obtained:

$$
\int_0^{\frac{\widehat{c_\alpha}}{\beta}} \frac{x^{\gamma-1}e^{-x}}{\Gamma(\gamma)} dx = 1 - \alpha, \tag{4}
$$

where $\Gamma(\cdot)$ denotes the Gamma function. Finally, independence is determined by comparing $T$ against $\widehat{c_\alpha}$.

For FUIT, the key distinction from KIT lies in its aggregation process: the cross-covariance matrix is computed through client-wise aggregation, $C_{xy} = \sum_k C_{xy}^{(k)}$ with total sample size $n = \sum_k n_k$. Here, $C_{xy}^{(k)} = \frac{1}{n}\phi(\boldsymbol{x}^{(k)})^\top \mathbf{H}_k^\top \mathbf{H}_k \phi(\boldsymbol{y}^{(k)})$, where $\phi(\boldsymbol{x}^{(k)}) \in \mathbb{R}^{n_k \times h}$ is obtained by applying the same random feature map to the local samples $\boldsymbol{x}^{(k)} := (\boldsymbol{x}_{k,1}, \boldsymbol{x}_{k,2}, \ldots, \boldsymbol{x}_{k,n_k}) \in \mathbb{R}^{d_x \times n_k}$ of client $k$. The local centering matrix is $\mathbf{H}_k = \mathbf{I} - \frac{1}{n_k}\mathbf{1}_{n_k}\mathbf{1}_{n_k}^\top$. The terms $\phi(\boldsymbol{y}^{(k)})$, $C_{xx}^{(k)}$, and $C_{yy}^{(k)}$ are defined analogously.

We now show that this aggregation is equivalent to applying the method to the concatenated client features. Define $f_k := \mathbf{H}_k \phi(\boldsymbol{x}^{(k)}) \in \mathbb{R}^{n_k \times h}$, $f_{con} := [f_1; f_2; \ldots; f_K] \in \mathbb{R}^{n \times h}$. Then $C_{xy} = \sum_k C_{xy}^{(k)} = \frac{1}{n}\sum_k f_k^T f_k = \frac{1}{n}f_{con}^T f_{con}$. Thus, the aggregated covariance is exactly the covariance computed on the concatenated features from all clients.

As discussed earlier, such naive aggregation is problematic under client heterogeneity, leading to uncontrolled Type I error or reduced test power when client-specific marginals or dependence structures differ. We therefore conclude that existing FedIT methods are inadequate for addressing these fundamental challenges, and introduce a new approach in the next section to overcome these limitations.

# 4. Methodology

In this section, we present our solutions to the previously analyzed challenges. The overall framework is illustrated in Fig. 2, which outlines the key steps. The core module consists of a copula transform to achieve marginal alignment, together with a Canonical Correlation Analysis (CCA, (Härdle & Simar, 2015)) with random projections to strengthen intra-client dependencies. These intra-client procedures, in turn, enable a more effective aggregation process, thereby enhancing the power of the global test. In what follows, we introduce each component of the framework in detail.

## 4.1. The Copula of Distributions

The copula (Nelsen, 2006) plays a central role in characterizing dependence among random variables. It fully captures the essential underlying dependence structure separately from the marginal distributions and is invariant under strictly increasing transformations of the marginal variables.

**Definition 4.1** (Copula and copula transformation (Sklar, 1959; Póczos et al., 2012)). Consider a $d$-dimensional random vector $\boldsymbol{X} = (X_1, \ldots, X_d)$ with continuous marginal cumulative distribution functions (cdfs) $F_i(x_i) = \mathbb{P}(X_i \leq x_i)$, $i \in [d]$. By Sklar's theorem, the joint cumulative distribution function can be uniquely represented as

$$F(\boldsymbol{x}) = \mathbb{P}(\boldsymbol{X} \leq \boldsymbol{x}) = \mathfrak{C}(F_1(x_1), \ldots, F_d(x_d)). \quad (5)$$

where $\mathfrak{C} : [0,1]^d \to [0,1]$ is the copula of $\boldsymbol{X}$. The corresponding copula transformation is the coordinate-wise marginal mapping

$$\boldsymbol{F}(\boldsymbol{x}) := (F_1(x_1), \ldots, F_d(x_d)). \quad (6)$$

Applying this copula transformation to $\boldsymbol{X}$ yields the transformed vector $\boldsymbol{U} := \boldsymbol{F}(\boldsymbol{X})$ in the copula space.

The probability integral transform shows that the copula transformation yields uniform marginals.

**Theorem 4.2** (Probability integral transform (Nelsen, 2006)). *For a univariate random variable $X$ with cdf $F$, $U := F(X)$ is uniformly distributed on $[0, 1]$.*

As a result, the $d$-dimensional transformed vector $\boldsymbol{U} = (U_1, \ldots, U_d)$ satisfies $U_i = F_i(X_i) \sim \text{Uniform}[0,1]$ for each $i \in [d]$, and its joint distribution is the copula $\mathfrak{C}$. Given a sample matrix $[x_{j,i}]_{n \times d}$ with $n$ samples, each marginal cdf $F_i$ can be estimated by the empirical marginal cdf $F_{n,i}(x) := \frac{1}{n} \sum_{j=1}^n \mathbb{I}[x_{j,i} \leq x]$, $i \in [d]$, where $\mathbb{I}$ is the indicator function. Then, for a $d$-dimensional vector $\boldsymbol{x}$, the empirical copula transformation is

$$\boldsymbol{F}_n(\boldsymbol{x}) := [F_{n,1}(x_1), F_{n,2}(x_2), ..., F_{n,d}(x_d)], \quad (7)$$

which converges uniformly to the true copula transformation as the sample size increases. The following theorem provides a formal characterization of this convergence.

**Theorem 4.3** (Convergence of the empirical copula (Póczos et al., 2012)). *Let $\boldsymbol{F}$ denote the copula transformation defined above, and let $\boldsymbol{F}_n$ be its empirical counterpart. Then,*

$$\mathbb{P}\left[ \sup_{\boldsymbol{x} \in \mathbb{R}^d} \|\boldsymbol{F}(\boldsymbol{x}) - \boldsymbol{F}_n(\boldsymbol{x})\|_2 > \epsilon \right] \leq 2d \exp\left( -\frac{2n\epsilon^2}{d} \right). \quad (8)$$

The exponential convergence rate with sample size supports the practical effectiveness of the copula transformation. Computing $\boldsymbol{F}_n(\boldsymbol{X})$ requires sorting each marginal of $\boldsymbol{X}$ over $n$ samples, resulting in $\mathcal{O}(dn \log n)$ operations.

## 4.2. Intra-Client Dependence Measurement

The goal of this section is to extract informative intra-client dependence signals, thereby enabling more effective cross-client aggregation. We build on the Hirschfeld Gebelein Rényi Maximum Correlation Coefficient (HGR) (Gebelein, 1941), which measures dependence as

$$\text{hgr}(X, Y) = \sup_{f,g} \rho(f(X), g(Y)), \quad (9)$$

where the supremum is taken over Borel-measurable functions $f$ and $g$ with finite variance, and $\rho$ denotes Pearson's correlation coefficient. Intuitively, HGR captures the maximal correlation achievable through nonlinear transformations of the variables. In practice, a common class of estimators approximates the transformation functions $f$ and $g$ using random projections, as proposed by (Lopez-Paz et al., 2013). This approach avoids explicit optimization over general function classes and leverages random-projection approximation properties, summarized below.

**Theorem 4.4** (Random-projection approximation (Rahimi & Recht, 2008)). *Let $p(w)$ be a distribution on $\Omega$ and assume that $\sup_{x,w} |\phi(x; w)| \leq 1$. Define $\mathcal{F} := \{ f(x) = \int_\Omega u(w)\phi(x; w)dw \mid |u(w)| \leq Cp(w) \}$. Draw $w_1, ..., w_h$ i.i.d from $p(w)$. Further let $\delta > 0$, and $c$ be some $L$-Lipschitz loss function, and consider data $\{x_i, o_i\}_{i=1}^n$ drawn i.i.d from some arbitrary $\mathbb{P}_{XO}$. The linear regression coefficient $u_1, ..., u_h$ defining $f_h(x) = \sum_{j=1}^h u_j \phi(x; w_j)$ are estimated by minimizing the empirical risk $c(f_h(x), o)$. The resulting estimator has a distance from the $c$-optimal estimator in $\mathcal{F}$ bounded by*

$$\mathbb{E}_{\mathbb{P}}[c(f_h(x), o)] - \min_{f \in \mathcal{F}} \mathbb{E}_{\mathbb{P}}[c(f(x), o)] \leq$$

$$\mathcal{O}\left( \left( \frac{1}{\sqrt{n}} + \frac{1}{\sqrt{h}} \right) LC \sqrt{\log \frac{1}{\delta}} \right) \quad (10)$$

*with probability at least $1 - 2\delta$.*

Intuitively, Theorem 4.4 shows that replacing optimized projection parameters with randomly sampled $\{w_i\}_{i=1}^h$ incurs only a bounded approximation error. Therefore, Eq. (9) can

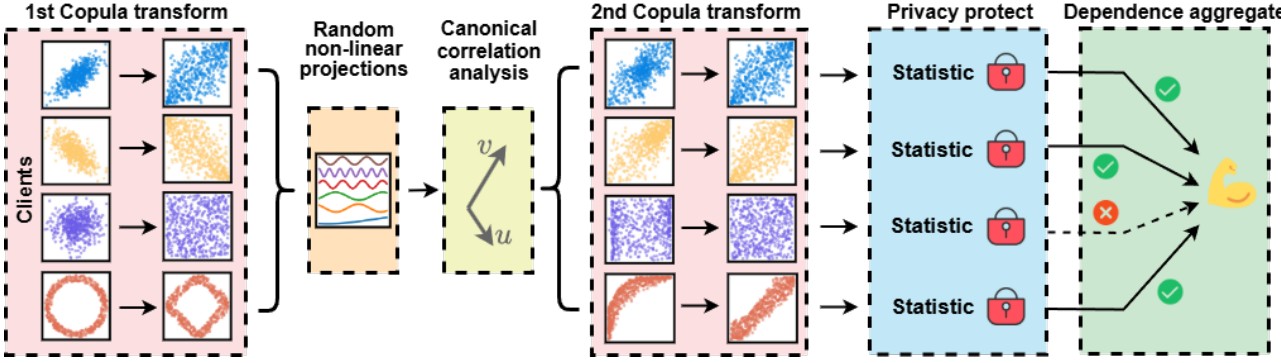

*Figure 2.* An overview of our FedIT-CS framework. Each client first applies an empirical copula transformation to ensure translation and scale invariance, followed by random feature projection to capture nonlinear dependence signals. Canonical correlation analysis is then used to linearly combine these features and maximize intra-client dependence strength, preparing informative signals for subsequent aggregation. To facilitate aggregation across heterogeneous clients, a second copula transformation aligns marginal distributions. A privacy-preserving module is applied before aggregation, and a subset of clients is selected to maximize the overall testing power.

be approximated by optimizing $\boldsymbol{u}$ and $\boldsymbol{v}$ to maximize the correlation between the projected features:

$$\max_{\boldsymbol{u},\boldsymbol{v}} \rho\left(\boldsymbol{u}^T\phi(\boldsymbol{x}), \boldsymbol{v}^T\phi(\boldsymbol{y})\right), \tag{11}$$

where $\phi(\boldsymbol{x})$ and $\phi(\boldsymbol{y})$ denote the random nonlinear projections of $\boldsymbol{x}$ and $\boldsymbol{y}$, respectively. The remaining task is to specify the nonlinear feature map and optimize $\boldsymbol{u}$ and $\boldsymbol{v}$.

In general, $\phi(\boldsymbol{x})$ can be instantiated using a broad class of randomized constructions, including random Fourier features, random neural features, and other randomized basis expansions. Following (Lopez-Paz et al., 2013), we adopt random Fourier features (RFFs) (Rahimi & Recht, 2007) for their computational efficiency and simplicity.

Formally, the weights are drawn as $w_i \sim \mathcal{N}(0, s\mathbf{I})$ and $b_i \sim$ Uniform$(-\pi, \pi)$ for $i \in [h]$, where $s$ is a tunable parameter, typically set empirically as a linear function of the input dimensionality. The corresponding RFF map is then defined by $\phi(\boldsymbol{x}) := \left[\cos(w_1^T\boldsymbol{x} + b_1); \ldots; \cos(w_h^T\boldsymbol{x} + b_h)\right]^T \in \mathbb{R}^{n \times h}$, and analogously for $\phi(\boldsymbol{y})$ throughout.

The optimization over $\boldsymbol{u}$ and $\boldsymbol{v}$ then reduces to a Canonical Correlation Analysis (CCA) problem (Härdle & Simar, 2015). Formally, let $C_{xy} := \mathrm{cov}(\phi(\boldsymbol{x}), \phi(\boldsymbol{y})) \in \mathbb{R}^{h \times h}$ and define $C_{xx}$ and $C_{yy}$ analogously. The leading canonical directions are obtained by solving

$$\begin{bmatrix} 0 & C_{xx}^{-1}C_{xy} \\ C_{yy}^{-1}C_{yx} & 0 \end{bmatrix} \begin{bmatrix} \boldsymbol{u} \\ \boldsymbol{v} \end{bmatrix} = \rho \begin{bmatrix} \boldsymbol{u} \\ \boldsymbol{v} \end{bmatrix}, \tag{12}$$

where the largest nonnegative eigenvalue corresponds to the leading canonical correlation $\rho_1$. As a result, after nonlinear feature projection and CCA optimization, the dominant potentially nonlinear dependence components are distilled into a low-dimensional representation, which facilitates subsequent aggregation. Specifically, the procedure outputs the

one-dimensional projected variables $\boldsymbol{u}^\top\phi(\boldsymbol{x}) \in \mathbb{R}^{1 \times n}$ and $\boldsymbol{v}^\top\phi(\boldsymbol{y}) \in \mathbb{R}^{1 \times n}$, whose correlation equals $\rho_1$.

*Remark.* This method is closely related to KIT analyzed in Sec. 3.2, as both are built upon $C_{xy}$. The key difference is that KIT considers all eigenvalues (via the Frobenius norm), while our approach focuses on the largest one (through CCA). In federated settings, this helps avoid eigenvalue cancellation and dependence dilution. For instance, in Sec. 3.2, $\Sigma_{1,xy} = 0.3$ and $\Sigma_{2,xy} = -0.3$ sum to zero, but according to Eq. (12), our method still retains the largest eigenvalue of 0.3 for both clients, providing consistent dependence signals. Moreover, the resulting low-dimensional output reduces communication cost and enhances privacy, as it makes the reconstruction of a client's raw data more difficult.

For subsequent aggregation, we apply the copula transformation again to further align the marginal distributions of $\boldsymbol{u}^T\phi(\boldsymbol{x})$ and $\boldsymbol{v}^T\phi(\boldsymbol{y})$. The transformed representations are denoted by $(\boldsymbol{r_x}, \boldsymbol{r_y})$, where $\boldsymbol{r_x}, \boldsymbol{r_y} \in \mathbb{R}^{1 \times n}$, with corresponding random variables $R_x$ and $R_y$. By construction, their marginal distributions satisfy $R_x, R_y \sim \mathrm{Uniform}(0, 1)$. We next describe the aggregation procedure in detail.

### 4.3. Stacking-based Aggregation Strategy

After the intra-client dependence modeling stage, each client $k \in [K]$ produces the copula-transformed outputs $(\boldsymbol{r}_x^{(k)}, \boldsymbol{r}_y^{(k)})$. We now describe how these outputs are aggregated across clients. For clarity, we first omit the client subset selection process (introduced later) and focus on the computation and communication of summary statistics and the server-side aggregation procedure.

**Computation of summary statistics.** Since nonlinear dependencies have already been captured within each client, inter-client aggregation only needs to summarize linear cor-

relation information. Let $\mathcal{I}$ denote the subset of selected clients and $n_{\mathcal{I}} = \sum_{k \in \mathcal{I}} n_k$ be the total sample size. Each client $k \in \mathcal{I}$ computes its local summary statistics:

$$
e_x^{(k)} = \sum_{i=1}^{n_k} r_{x;i}^{(k)}, \; e_y^{(k)} = \sum_{i=1}^{n_k} r_{y;i}^{(k)}, \; m_{xy}^{(k)} = \sum_{i=1}^{n_k} r_{x;i}^{(k)} r_{y;i}^{(k)},
$$

$$
m_{xx}^{(k)} = \sum_{i=1}^{n_k} r_{x;i}^{(k)} r_{x;i}^{(k)}, \; m_{yy}^{(k)} = \sum_{i=1}^{n_k} r_{y;i}^{(k)} r_{y;i}^{(k)}.
$$

$$(13)$$

These local statistics are then aggregated by the server to obtain the global summary quantities:

$$
e_x^{\mathcal{I}} = \sum_{k \in \mathcal{I}} e_x^{(k)}, \; e_y^{\mathcal{I}} = \sum_{k \in \mathcal{I}} e_y^{(k)}, \; m_{xy}^{\mathcal{I}} = \sum_{k \in \mathcal{I}} m_{xy}^{(k)},
$$

$$
m_{xx}^{\mathcal{I}} = \sum_{k \in \mathcal{I}} m_{xx}^{(k)}, \; m_{yy}^{\mathcal{I}} = \sum_{k \in \mathcal{I}} m_{yy}^{(k)}.
$$

$$(14)$$

Based on these aggregated quantities, the global correlation coefficient is computed as

$$
\rho_{xy}^{\mathcal{I}} = \frac{n_{\mathcal{I}} m_{xy}^{\mathcal{I}} - e_x^{\mathcal{I}} e_y^{\mathcal{I}}}{\sqrt{n_{\mathcal{I}} m_{xx}^{\mathcal{I}} - (e_x^{\mathcal{I}})^2} \sqrt{n_{\mathcal{I}} m_{yy}^{\mathcal{I}} - (e_y^{\mathcal{I}})^2}}. \quad (15)
$$

Importantly, this computation requires only a small set of aggregated summary statistics rather than direct access to raw client outputs. We next explain how these statistics are securely combined under privacy protection.

**Privacy-preserving aggregation.** Although the transmitted one-dimensional statistics are already highly compressed, making reconstruction of the original client data difficult, we further employ additive Homomorphic Encryption (HE) (Paillier, 1999) to provide stronger privacy protection. Importantly, this choice is enabled by our preceding dependence modeling step: after nonlinear dependence is distilled into one-dimensional projected variables, cross-client aggregation reduces to summing a small set of additive summary statistics for computing linear correlation, which can be handled naturally by additive HE. To implement this privacy-preserving aggregation, a trusted authority generates a shared public–private key pair $(pk, sk)$. All clients encrypt their local quantities using $pk$, while $sk$ is held by a designated decrypting client or the trusted authority. Under the standard honest-but-curious setting, the server follows the protocol but may attempt to infer private information from transmitted messages; however, it cannot decrypt any client-level statistics. Thus, client $k$ transmits only the encrypted statistics, $Enc(n_k)$, $Enc(e_x^{(k)})$, $Enc(e_y^{(k)})$, $Enc(m_{xx}^{(k)})$, $Enc(m_{xy}^{(k)})$, and $Enc(m_{yy}^{(k)})$, rather than their plaintext values. Since all ciphertexts are generated under the same shared key pair, the server can directly homomorphically aggregate them to obtain encrypted global summaries such as $Enc(e_x^{\mathcal{I}}) = \sum_{k \in \mathcal{I}} Enc(e_x^{(k)})$ and

$Enc(m_{xy}^{\mathcal{I}}) = \sum_{k \in \mathcal{I}} Enc(m_{xy}^{(k)})$, with the remaining terms handled analogously in the same manner.

The aggregated ciphertexts are then sent to the designated decrypting party, which recovers only the global statistics $n_{\mathcal{I}}, e_x^{\mathcal{I}}, e_y^{\mathcal{I}}, m_{xx}^{\mathcal{I}}, m_{xy}^{\mathcal{I}}$, and $m_{yy}^{\mathcal{I}}$, computes $\rho_{xy}^{\mathcal{I}}$, and returns the final result to the server. This additional communication round ensures that the server never accesses plaintext client-level statistics. More precisely, HE reveals only aggregated summaries, while individual client submissions remain encrypted throughout the protocol. Although the decrypting party observes the aggregated quantities, it does not access the plaintext statistics submitted by individual clients. Moreover, because each client communicates only a small number of one-dimensional summary statistics, the additional HE overhead remains modest in our framework. Further discussion of privacy-preserving techniques is provided in Appendix B, while the complete HE protocol, its interpretation, and practical communication and computation overhead are detailed in Appendix D.

**Client subset selection strategy.** The remaining problem is how to select a subset of clients that yields a stronger global dependence signal and, consequently, higher testing power. Rather than relying on computationally expensive permutation procedures (Good, 2013), we use the aggregated summary correlation coefficient $\rho_{xy}^{\mathcal{I}}$ in Eq. (15) as a practical criterion. This quantity is computed directly from client-side summary statistics and measures the dependence strength retained after aggregation. We therefore use it as a surrogate power score for comparing candidate client subsets. Our experiments confirm that this criterion is effective in practice (see Appendix I.3). This naturally induces a subset selection problem over client combinations.

A straightforward strategy is to enumerate all nonempty client subsets, evaluate their corresponding scores $\rho_{xy}^{\mathcal{I}}$, and select the subset with the largest value. This yields an exact discrete selection strategy, but becomes computationally intractable as the number of clients $K$ increases, since the search space contains $2^K - 1$ candidate subsets. To improve scalability, we further introduce a soft relaxation that converts subset selection into a continuous optimization problem. Specifically, each client $k \in [K]$ is assigned a learnable weight $p_k \in [0, 1]$. Let $n_{\mathcal{P}} = \sum_{k \in [K]} p_k n_k$, and define the weighted aggregated statistics analogously to Eq. (15); for example,

$$
e_x^{\mathcal{P}} = \sum_{k \in [K]} p_k e_x^{(k)}, \; m_{xy}^{\mathcal{P}} = \sum_{k \in [K]} p_k m_{xy}^{(k)},
$$

$$
\rho_{xy}^{\mathcal{P}} = \frac{n_{\mathcal{P}} m_{xy}^{\mathcal{P}} - e_x^{\mathcal{P}} e_y^{\mathcal{P}}}{\sqrt{n_{\mathcal{P}} m_{xx}^{\mathcal{P}} - (e_x^{\mathcal{P}})^2} \sqrt{n_{\mathcal{P}} m_{yy}^{\mathcal{P}} - (e_y^{\mathcal{P}})^2}}.
$$

$$(16)$$

This relaxed formulation can be efficiently optimized using standard gradient-based methods in practice. Compared

*Table 1.* Comparison of FUIT (Li et al., 2024) and the variants of our proposed FedIT-CS.

| Method | FUIT | FedIT-CS-S (Ours) | FedIT-CS-M (Ours) | FedIT-CS-ML (Ours) |
|---|---|---|---|---|
| Theoretical soundness | ✗ | ✗ | ✓ | ✓ |
| Privacy protection | ✗ | ✓ | ✓ | ✓ |
| Maximum power selection | ✗ | ✗ | ✓ | ✓ |
| Local computation cost | $\mathcal{O}(nh^2)$ | $\mathcal{O}(Bn\log n + Bnh^2)$ | $\mathcal{O}(Bn\log n + Bnh^2)$ | $\mathcal{O}(Bn\log n + Bnh^2)$ |
| Aggregation cost | $\mathcal{O}(Kh^2)$ | $\mathcal{O}(KB)$ | $\mathcal{O}(2^K B)$ | $\mathcal{O}(KB)$ |
| Communication cost | $\mathcal{O}(Kh^2)$ | $\mathcal{O}(KB)$ | $\mathcal{O}(KB)$ | $\mathcal{O}(KB)$ |

with the discrete case, its privacy-preserving implementation requires additional care: HE can still support gradient computation for the aggregated quantities with respect to $p_k$, enabling each client to privately update its local weight parameter $p_k$ (please see Appendix D for details).

### 4.4. The Overall Framework

Above, we have detailed the intra-client modules and the aggregation procedure. We now present the complete framework, termed **FedIT-CS** (Federated Independence Testing with Copula Alignment and Stacking Aggregation). Depending on the aggregation strategy, we consider two variants: **FedIT-CS-M**, which performs maximum-power selection over client subsets, and **FedIT-CS-ML**, which further generalizes the framework through mixed linear aggregation while achieving linear complexity in the number of clients.

**Permutation-based testing.** To obtain $p$-value for hypothesis testing, we adopt a permutation-based procedure (Good, 2013). Formally, we generate $B$ permuted samples $(\sigma_t(\boldsymbol{x}), \boldsymbol{y})$ with $t \in [B]$, where each $\sigma_t$ is an independent derangement. These permuted pairs simulate samples under $\mathcal{H}_0$. Each client can perform this procedure independently, producing local null samples which, after intra-client transformation obtain copula output, are aggregated to compute the global test statistic. This enables an approximation of the null distribution of the aggregated test.

**Sample splitting.** A key detail is that we cannot use the same data both for learning the aggregation strategy and for performing the test, as this would invalidate Type I error control. To address this, we adopt a straightforward data-splitting strategy (Jitkrittum et al., 2017; Liu et al., 2020), which is simple and direct but inevitably reduces statistical power. Actually, more advanced strategies such as (Schrab et al., 2022; Kübler et al., 2020) could be considered, and we leave this as an important direction for further work.

For comparison with FUIT, we also introduce a naive variant, **FedIT-CS-S**, which directly pools copula outputs to compute covariance. This is equivalent to applying FedIT-CS-M to the full client set without subset selection. FedIT-CS-S

does not require sample splitting, but it sacrifices theoretical guarantees. Table 1 summarizes all variants of our framework. It shows that only FedIT-CS-M and FedIT-CS-ML achieve both *theoretical soundness* and *maximum-power selection*, while all FedIT-CS variants provide *privacy protection*. In terms of computational cost, FedIT-CS-S and FedIT-CS-ML support efficient aggregation with complexity $\mathcal{O}(KB)$, whereas FedIT-CS-M incurs exponential complexity $\mathcal{O}(2^K B)$ due to exhaustive subset enumeration.

---

**Algorithm 1** FedIT-CS Framework

---

**Input:** Federal samples $\boldsymbol{Z}_k = \{(x_i^{(k)}, y_i^{(k)})\}_{i=1}^{n_k}, k \in [K]$, the number of feature sampling $h$, significance level $\alpha$, the number of permutation $B$.

**Output:** $X \perp\!\!\!\perp Y$ or $X \not\!\perp\!\!\!\perp Y$.

1: Split client data into train/ test sets: $\boldsymbol{Z}_k = \boldsymbol{Z}_k^{tr} \cup \boldsymbol{Z}_k^{te}$.
2: ◁ **Intra-client dependency modeling with $\boldsymbol{Z}_k^{tr}$.**
3: Apply copula transform, random projections,
4: Calculate CCA as in Eq. (12).
5: ◁ **Subset selection with $\boldsymbol{Z}_k^{tr}, k \in [K]$.**
6: Apply the second copula transform.
7: Aggregate with Eqs. (15) or (16).
8: Obtain the aggregation weights $\boldsymbol{p}^* = (p_1^*, p_2^*, \ldots, p_K^*)$.
9: ◁ **Permutation test with $\boldsymbol{p}^*$ on $\boldsymbol{Z}_k^{te}, k \in [K]$.**
10: Generate $B$ permuted samples $(\sigma_t(\boldsymbol{x}), \boldsymbol{y}), \ t \in [B]$,
11: Each perform intra-client transform.
12: Compute statistic sequence $\{\rho_{xy}^{\mathcal{P}}, \rho_{xy}^{\mathcal{P},\sigma_1}, ..., \rho_{xy}^{\mathcal{P},\sigma_B}\}$.
13: Obtain $p$-value$= [1 + \sum_{t=1}^B \mathbb{I}\{\rho_{xy}^{\mathcal{P},\sigma_t} \geq \rho_{xy}^{\mathcal{P}}\}]/[1 + B]$.
14: Reject $X \perp\!\!\!\perp Y$ if $p$-value $\leq \alpha$; otherwise accept.

---

**Algorithm.** The overall procedure is summarized in Algorithm 1. As a preprocessing step, each client splits its data into training $\boldsymbol{Z}_k^{tr}$ and testing $\boldsymbol{Z}_k^{te}$ (Line 1). The test consists of two phases: (i) intra-client dependency modeling and client subset selection using the training data (Lines 2–8), and (ii) a permutation test with the learned aggregation weights to perform inference on the testing data (Lines 9–14). The computational complexity of different variants is summarized in Table 1, and compatible privacy-preserving schemes can be applied accordingly.

## 5. Theoretical Analysis

Let the aggregation algorithm be denoted by $\mathcal{A}$, which determines the client weights $p_k$, $k \in [K]$. When $p_k \in \{0, 1\}$, the resulting aggregation corresponds to the FedIT-CS-M strategy; when $p_k \in [0, 1]$, it corresponds to FedIT-CS-ML. We first study an oracle setting in which $\mathcal{A}$ attains the theoretically optimal aggregation weights. In the asymptotic regime where both the sample size and the number of random features are sufficiently large, let the resulting aggregated coefficient be denoted by $\rho_{\mathcal{A}}$. The following theorem establishes its population-level soundness.

**Theorem 5.1** (Soundness of aggregated statistics). *Let $\rho_{\mathcal{A}}$ denote the aggregated coefficient obtained by the aggregation algorithm $\mathcal{A}$. Assume $\mathcal{A}$ is theoretically optimal. Then, under the null hypothesis $\mathcal{H}_0$, we have $\rho_{\mathcal{A}} = 0$, whereas under the alternative hypothesis $\mathcal{H}_1$, we have $\rho_{\mathcal{A}} > 0$.*

*Proof.* The proof is given in Appendix F. □

*Remark.* Theorem 5.1 is established under an oracle aggregation setting and therefore serves as a population-level characterization of the proposed aggregation principle. For the implemented finite-sample procedures, FedIT-CS-ML involves optimizing a generally non-convex objective, for which deriving a complete finite-sample characterization of adaptive optimization dynamics is challenging (Reddi et al., 2019). We therefore focus here on the soundness of the target aggregation mechanism itself. In practice, exact recovery of the global optimum is not necessary for the method to be useful; it suffices that the learned weights enhance the aggregated dependence signal, which is consistently supported by our empirical results.

The approximation components are controlled separately. By Theorems 4.3 and 4.4, the empirical copula approximation error and random-feature approximation error vanish at rates $\mathcal{O}(1/\sqrt{n})$ and $\mathcal{O}(1/\sqrt{h})$, respectively, with high probability. These results support the practical relevance of the asymptotic analysis when the sample size and feature dimension are sufficiently large. We next establish a finite-sample Type I error bound for our test.

**Theorem 5.2** (Type I error bound). *Assume the null hypothesis $\mathcal{H}_0$ is true. For any significance level $\alpha \in (0, 1)$, the bound for the Type I error is given by*

$$\mathbb{P}(\text{p-value} \leq \alpha | \mathcal{H}_0) \leq \alpha. \tag{17}$$

*Proof.* The proof is given in Appendix G. □

This guarantees the validity of our test. Together with the theoretical soundness of the aggregated statistic, we thus establish the foundation of our method's reliability.

## 6. Performance Evaluation

We compare the following tests: FUIT (Li et al., 2024), FedIT-CS-S, FedIT-CS-M, and FedIT-CS-ML. To further assess the potential statistical power loss caused by sample splitting, we additionally include two variants, FedIT-CS-M-F and FedIT-CS-ML-F, in which the aggregation strategy is trained on extra data not used for testing. For fairness, all methods use $h = 10$ random features, significance level $\alpha = 0.05$, and split ratio 0.2. We evaluate them on both synthetic and real-world datasets. Detailed experimental settings and additional results are provided in Appendices H and I, including further synthetic studies, comparisons with additional baselines, results on an additional real-world fMRI hippocampus dataset, evaluations with more clients, and runtime comparisons. Our code is available at `https://github.com/lhq12/FedIT`.

### 6.1. Results on Synthetic Datasets

We first evaluate the proposed methods under controlled synthetic settings with heterogeneous clients.

**Experimental settings.** We consider three heterogeneous client scenarios. (i) *Covariance (Cov)*: linear dependency with client-specific correlation coefficients 0.5, -0.5, and 0.02; the sample size ratio $n_1 : n_2 : n_3 = 1 : 1 : 2$, with $n_1$ ranging from 100 to 400. (ii) *Frequency (Freq)*: a sinusoid model $(X, Y) \propto 1 + \sin(\omega x)\sin(\omega y)$ with $\omega = 2, 3, 4$ across clients; the sample size ratio $n_1 : n_2 : n_3 = 1 : 1 : 1$, with $n_1$ from 300 to 900. (iii) *Functional (Func)*: nonlinear relations $Y = f(X) + \epsilon$, where $f \in \{\sin(\cdot), \cos(\cdot), (\cdot)^2\}$ and $\epsilon \sim \mathcal{N}(0, 1)$; the sample size ratio $n_1 : n_2 : n_3 = 4 : 2 : 1$, with $n_1$ from 400 to 1600. Type I error rates are evaluated using permuted samples. Each point is averaged over 100 randomized trials, and all figures are plotted against $n_1$.

**Performance and analysis.** Results are presented in Fig. 3. All methods generally control Type I error, with slight deviations in some settings. On both the Covariance and Frequency settings, our methods consistently outperform FUIT, which can be attributed to the effectiveness of our aggregation strategy. Comparing the variants with and without additional training data (-F vs. non -F), we observe that data splitting indeed reduces statistical power, though FedIT-CS-ML still achieves better performance than FUIT. Interestingly, despite being designed primarily for efficiency, our linear-time variant FedIT-CS-ML outperforms FedIT-CS-M. This improvement may stem from the greater flexibility of its solution space, which provides additional practical benefits to strengthen the dependency signal.

### 6.2. Results on Real-World Data

We further evaluate our methods in a real-world heterogeneous setting under distribution shifts.

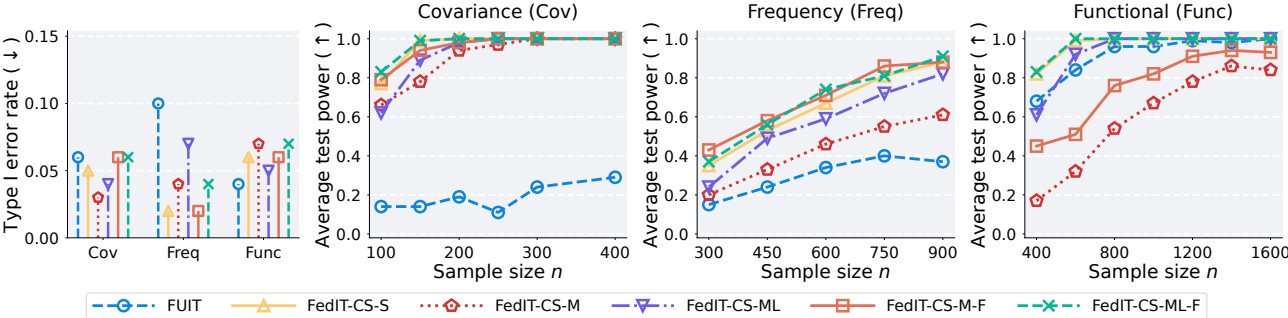

*Figure 3.* Performance comparison on three synthetic datasets under heterogeneous client settings. Left: average Type I error rate. The remaining three panels report the average test power on the Covariance, Frequency, and Functional datasets, respectively.

**Experimental settings.** We further evaluate our methods on the Sachs dataset (Sachs et al., 2005), which records 11 phosphorylated proteins and phospholipids in primary immune cells under multiple stimulatory and inhibitory interventions. We treat seven experimental conditions as distinct clients. A common reference signaling graph derived from interventional experiments is available, making this dataset suitable for evaluating aggregation across heterogeneous client distributions without assuming identical observed marginals. Visualizations of client-wise distributions are provided in Appendix H.2. The reference network contains 11 nodes, yielding 55 variable pairs: 18 independent pairs for Type I error evaluation and 37 dependent pairs for Type II error evaluation. In each run, we randomly select 3 of the 7 clients, evaluate all pairs, and report the average performance. This procedure is repeated 50 times with independently sampled client subsets.

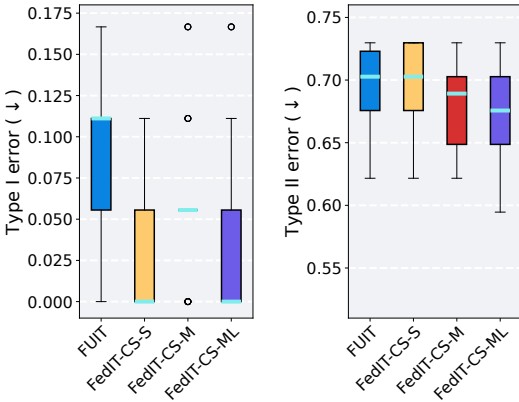

*Figure 4.* The results on Sachs dataset.

**Performance and analysis.** Results are shown in Fig. 4. Compared with FUIT, all variants of our method achieve tighter Type I error control while simultaneously reducing Type II error, indicating stronger overall detection power. These findings provide empirical support for the advantages of our framework in realistic heterogeneous settings.

## 7. Conclusion

This paper presents FedIT-CS, a federated independence testing framework that addresses both theoretical and practical challenges under client heterogeneity. Building on our analysis of existing limitations, FedIT-CS combines copula-based marginal alignment with stacking-based aggregation to improve test validity and statistical power, while also supporting efficient and privacy-preserving implementations. Theoretical results further establish the soundness and reliability of the proposed framework. Extensive experiments demonstrate its clear advantages over prior approaches. An interesting direction for future work is to extend FedIT-CS to conditional independence testing task.

*Discussion.* Our problem formulation assumes a shared independence status across clients, as in related federated causality studies. To examine the scope of this assumption, we briefly explore extended settings, with results in Appendix J. From a methodological perspective, eliminating sample splitting and developing adaptive bandwidth selection are also promising directions for further improving the overall testing power and practical performance.

## Acknowledgements

Yixin Ren, Hongquan Liu, Yewei Xia, and Shuigeng Zhou were supported by National Natural Science Foundation of China (NSFC) (No. 62372116), Jihong Guan was supported by NSFC (No. 62372326), Yixin Ren was also supported by Tencent Rhino-Bird Elite Talent Program. The computations in this research were partially performed using the CFFF platform of Fudan University. The authors would like to thank the anonymous reviewers for their valuable advice.

## Impact Statement

This paper presents work whose goal is to advance the field of Machine Learning and Causality. There are many potential societal consequences of our work, none which we feel must be specifically highlighted here.

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

# Appendix Organization

## A. List of Symbols and Notations

| | |
|---|---|
| $\mathcal{O}, o$ | big, small O notion |
| $i.i.d.$ | independent and identically distributed |
| $\mathbb{R}$ | the set of real numbers |
| $\mathcal{B}(\mathbb{R})$ | Borel $\sigma$-algebra on $\mathbb{R}$ |
| $\mathbb{P}_X$ | marginal distribution of $X$ |
| $\mathbb{P}_{XY}$ | joint distribution of $X, Y$ |
| $F_X$ | cumulative distribution function (cdf) of $X$ |
| $\mathbb{E}[X]$ | expectation of $X$ |
| $\mathrm{Var}(X)$ | variance of $X$ |
| $\mathrm{Cov}(X, Y)$ | covariance of $X, Y$ |
| $X \perp\!\!\!\perp Y$ | random variables $X, Y$ are independent |
| $X \not\!\perp\!\!\!\perp Y$ | random variables $X, Y$ are not independent |
| $\mathrm{Tr}(\cdot)$ | the trace of a square matrix |
| $\mathbf{1}$ | an vector of all ones |
| $\mathbf{H}$ | centering matrix define as $\mathbf{H} = \mathbf{I} - \frac{1}{n}\mathbf{1}\mathbf{1}^T$ |
| $\odot$ | element-wise product |
| $[n]$ | denotes the set $\{1, 2, ..., n\}$ |
| $\rho$ | Pearson's correlation coefficient |
| $\Gamma(\cdot)$ | Gamma function |
| $\times$ | the product symbol of topological space |
| $\|\cdot\|_2$ | spectral norm |
| $\|\cdot\|_F$ | Frobenius norm |
| $\overset{d}{=}$ | equality in distribution |

## B. Related Work

**Centralized independence testing.** Traditional correlation-based measures, such as Pearson's coefficient (Benesty et al., 2009) and Kendall's $\tau$, capture only linear or monotonic associations and thus fail to detect general nonlinear dependencies. To characterize more complex relationships, including those arising in high-dimensional settings (Liu et al., 2022; Zhang et al., 2023c; Zhang & Zhu, 2023), a broad range of nonlinear dependence measures has been developed. These methods can be roughly grouped into three categories. (i) *Rank-based methods.* Chatterjee (2021) extends classical correlation ideas through rank statistics, yielding robustness to outliers and invariance under monotone transformations. (ii) *Distance-based methods.* Representative examples include distance covariance and related criteria (Székely et al., 2007; Lyons, 2013; Székely & Rizzo, 2013; Ren et al., 2023), which quantify dependence through characteristic functions and are effective for general nonlinear relationships. (iii) *Kernel-based methods.* This family defines dependence measures through cross-covariance

operators in reproducing kernel Hilbert spaces (RKHS). Early examples include Kernel Canonical Correlation (KCC) (Bach & Jordan, 2002), which maximizes the correlation between feature maps, and Constrained Covariance (COCO) (Gretton et al., 2005), which removes normalization constraints. The most widely used kernel-based approach is the Hilbert–Schmidt Independence Criterion (HSIC) (Gretton et al., 2007), which measures dependence via the squared Hilbert–Schmidt norm of the cross-covariance operator. Subsequent work has improved HSIC by accelerating computation for large-scale data (Zhang et al., 2018) and by optimizing kernel parameters to enhance test power (Jitkrittum et al., 2017; Ren et al., 2024; Xu et al., 2025; Ren et al., 2025b). Beyond these families, copula-based methods (Bouezmarni et al., 2012; Genest & Verret, 2005) provide a complementary centralized view by separating marginals from dependence structures, and can be combined with tools such as kernels for flexible dependence measures (Póczos et al., 2012).

These non-federated methods provide essential foundations for modern independence testing, but they are not designed for heterogeneous multi-client settings. In particular, they do not address how client-specific marginal shifts and heterogeneous dependence patterns affect cross-client aggregation, which is the central challenge studied in our federated formulation.

**Federated causal discovery.** Unlike traditional causal discovery methods (Spirtes, 2001; Yu et al., 2019; Ng et al., 2024), which typically assume independent and identically distributed ($i.i.d.$) data, federated causal discovery (FCD) must operate on decentralized and often heterogeneous data (Zhou et al., 2022). Existing FCD methods can be broadly grouped into three categories. (i) *Score-based methods*, which evaluate candidate graphs using predefined scoring functions (Huang et al., 2018; Ren et al., 2025a) together with search strategies (Tsamardinos et al., 2006; Chickering, 2003; 2020). For example, DARLIS (Ye et al., 2024) employs distributed simulated annealing, while PERI (Mian et al., 2023) builds on local GES (Chickering, 2003) with worst-case regret aggregation. (ii) *Continuous optimization-based methods*, which formulate structure learning as an optimization problem. NOTEARS (Zheng et al., 2018) pioneered this direction in the centralized setting, and federated extensions include NOTEARS-ADMM (Ng & Zhang, 2022), FedDAG (Gao et al., 2023), and Fed-CDI (Abyaneh et al., 2022). (iii) *Constraint-based methods*, which rely on conditional independence tests (Zhang et al., 2012; 2017; Pogodin et al., 2024) and graph-search procedures such as Peter–Clark (PC) (Spirtes et al., 2000). In the federated setting, FedPC (Huang et al., 2023) aggregates skeletons and orientations by voting under homogeneous data, while FedCDH (Li et al., 2024) introduces a federated conditional independence test (FCIT) and a federated independent change principle (FICP) to address heterogeneity, achieving strong performance in FCD.

Despite this progress, federated independence testing (FedIT) itself remains comparatively underexplored, even though it serves as a basic statistical primitive for federated causal discovery. In particular, the federated independence test used in FedCDH, referred to as FUIT, adopts a naive aggregation strategy in feature space. In this work, we revisit FUIT and show that such aggregation suffers from fundamental theoretical limitations and substantial power degradation under client heterogeneity. We then develop a new framework that directly addresses these core challenges.

**Privacy-preserving hypothesis testing.** Privacy-preserving statistical testing has received increasing attention. (i) In the non-federated setting, many studies develop differentially private (DP) techniques (Dwork & Roth, 2014; Mironov, 2017). For example, Kazan et al. (2023) proposed a black-box framework for privatizing general hypothesis tests; Priv-PC (Wang et al., 2020) designed DP algorithms for discrete data via sensitivity analysis of conditional Kendall's $\tau$ and Spearman's $\rho$, later extended to numerical data (Zhang et al., 2023a); Kusner et al. (2016) analyzed the sensitivity of HSIC; Kalemaj et al. (2024) added Laplace noise to regression residuals for conditional independence testing; and Kim & Schrab (2023) studied DP permutation tests for kernel-based methods to reduce privacy-induced power loss. Homomorphic encryption has also been explored: for instance, Lauter et al. (2015) proposed a private $\chi^2$ test for independence testing. (ii) In the federated setting, Pang et al. (2023) developed a secure federated correlation test (FED-$\chi^2$) and entropy estimator by reformulating the target computations as frequency-moment estimation and enabling secure aggregation through stable projections.

Overall, existing approaches are dominated by differential privacy, while homomorphic-encryption-based methods remain limited, especially beyond discrete settings such as the $\chi^2$ test. By contrast, our framework addresses continuous and nonlinear FedIT scenarios. Through client-side nonlinear transformation, it reduces cross-client aggregation to additive second-order summary statistics, naturally aligning with homomorphic encryption and enabling exact encrypted aggregation.

*Remark.* Although federated learning under distribution shift, such as covariate or label shift, has been extensively studied (Ramezani-Kebrya et al.; Wu et al., 2025), this literature primarily focuses on federated risk minimization, namely, learning predictive models that generalize across heterogeneous clients. In contrast, we study *federated independence testing*, a hypothesis-testing problem that determines whether $X \perp\!\!\!\perp Y$, rather than a prediction task. Our framework does not rely on a specific shift model; instead, it aims to detect dependence relationships under general and potentially unknown client heterogeneity, while ensuring valid Type I error control and maintaining strong testing power.

## C. Details of FedIT Methods

In this section, we provide a detailed explanation of the comparison table presented in Sec. 4.4 of the main paper, clarifying the theoretical properties, privacy guarantees, power-selection capability, and computational and communication costs of each method. For completeness and ease of reference, we reproduce the table below.

*Table 2.* Comparison of FUIT (Li et al., 2024) and the variants of our proposed FedIT-CS.

| Method | FUIT | FedIT-CS-S (Ours) | FedIT-CS-M (Ours) | FedIT-CS-ML (Ours) |
|---|---|---|---|---|
| Theoretical soundness | ✗ | ✗ | ✓ | ✓ |
| Privacy protection | ✗ | ✓ | ✓ | ✓ |
| Maximum power selection | ✗ | ✗ | ✓ | ✓ |
| Local computation cost | $\mathcal{O}(nh^2)$ | $\mathcal{O}(Bn\log n + Bnh^2)$ | $\mathcal{O}(Bn\log n + Bnh^2)$ | $\mathcal{O}(Bn\log n + Bnh^2)$ |
| Aggregation cost | $\mathcal{O}(Kh^2)$ | $\mathcal{O}(KB)$ | $\mathcal{O}(2^K B)$ | $\mathcal{O}(KB)$ |
| Communication cost | $\mathcal{O}(Kh^2)$ | $\mathcal{O}(KB)$ | $\mathcal{O}(KB)$ | $\mathcal{O}(KB)$ |

The entries in Table 2 are justified below in terms of their main properties and costs:

- **FUIT** (Li et al., 2024): FUIT directly aggregates local covariance matrices across all clients. As discussed in Sec. 3.2, this naive aggregation does not provide theoretical guarantees under client heterogeneity. Moreover, it includes neither privacy protection nor a client selection mechanism. Its local computation cost comes from constructing the local covariance matrix $C_{xy}^{(k)} \in \mathbb{R}^{h \times h}$ from random features of dimension $n \times h$, resulting in $\mathcal{O}(nh^2)$. Since aggregation only sums $C_{xy}^{(k)}$ over $k \in [K]$, both aggregation and communication costs are $\mathcal{O}(Kh^2)$.

- **FedIT-CS-S (Ours)**: FedIT-CS-S is the naive stacking variant, which directly aggregates all local second-order summary statistics, equivalently pooling the copula-transformed outputs. Hence, it does not enjoy theoretical soundness or perform client selection. However, because only encrypted summary statistics are transmitted, privacy protection can be incorporated as described in Appendix D. Its local computation includes empirical copula transformation via marginal sorting, $\mathcal{O}(n\log n)$, random-feature covariance computation, $\mathcal{O}(nh^2)$, and CCA, $\mathcal{O}(h^3)$. Since this procedure is repeated over $B$ permutation samples, the overall local cost is summarized as $\mathcal{O}(Bn\log n + Bnh^2)$. Aggregating the client-level summary statistics costs $\mathcal{O}(KB)$, and the communication cost is likewise $\mathcal{O}(KB)$.

- **FedIT-CS-M (Ours)**: FedIT-CS-M extends FedIT-CS-S by performing maximum-power selection over client subsets. It evaluates candidate subsets using the aggregated correlation statistic and selects the subset that yields the strongest global dependence signal. Its theoretical soundness follows from Theorem F.1, and privacy protection is supported as discussed in Appendix D. The local computation cost is the same as FedIT-CS-S, namely $\mathcal{O}(Bn\log n + Bnh^2)$. The main additional cost arises from exhaustive subset enumeration: evaluating all possible client subsets leads to aggregation complexity $\mathcal{O}(2^K B)$. The communication cost remains $\mathcal{O}(KB)$.

- **FedIT-CS-ML (Ours)**: FedIT-CS-ML further generalizes subset selection by allowing mixed linear aggregation weights over clients. This continuous relaxation provides a flexible mechanism for power maximization while avoiding exhaustive subset enumeration. Theoretical soundness is established in Theorem F.1, and privacy protection follows Appendix D. Its local computation cost remains $\mathcal{O}(Bn\log n + Bnh^2)$. Unlike FedIT-CS-M, the aggregation cost is reduced to $\mathcal{O}(KB)$ because the optimization scales linearly with the number of clients. The communication cost is also $\mathcal{O}(KB)$.

In addition, we consider several auxiliary variants in the experiments: **FedIT-CS-M-F**, **FedIT-CS-ML-F**, and **FedIT-CS-MB**. The **-F** variants use extra data to train the aggregation strategy separately, allowing us to more directly assess the statistical power loss introduced by sample splitting in FedIT-CS-M and FedIT-CS-ML.

**FedIT-CS-MB**, which is omitted from the main paper for brevity, replaces our correlation-based power score with a more computationally intensive permutation-based alternative. As discussed in Sec. 4.3 and further validated in Appendix I.3, the aggregated correlation coefficient already provides an efficient and informative proxy for dependence strength. These auxiliary variants are included for broader experimental comparison; detailed results are reported in Appendix I.

## D. Details of the Homomorphic Encryption Procedure

In the main paper, we briefly introduce Homomorphic Encryption (HE) as the privacy-preserving mechanism in FedIT-CS. Here, we provide a more detailed description of the protocol. HE enables computation directly on encrypted data, such that decryption yields exactly the result of the corresponding plaintext computation. This makes it well suited to privacy-sensitive aggregation tasks. Unlike differential privacy, HE introduces no statistical noise and preserves the exact target computation.

HE is particularly suitable for FedIT-CS because, after the client-side nonlinear dependence modeling stage, cross-client aggregation reduces to summing a small number of low-dimensional summary statistics. Consequently, the required encrypted operations remain simple, and the additional computational overhead is relatively modest.

We use additive homomorphic encryption, instantiated by the Paillier cryptosystem (Paillier, 1999), since aggregation only requires additive operations over summary statistics for computing linear correlation. Let $(pk, sk)$ be a public–private key pair, with $Enc(\cdot, pk)$ and $Dec(\cdot, sk)$ denoting encryption and decryption. For any plaintext values $x$ and $y$, additive homomorphism guarantees that

$$x + y = Dec(Enc(x, pk) \star Enc(y, pk), sk), \tag{18}$$

where $\star$ denotes addition in the ciphertext domain. Below, we explain how this property is used in FedIT-CS. Before aggregation, each client $k \in [K]$ produces the copula-transformed outputs $(r_x^{(k)}, r_y^{(k)})$ from the intra-client dependence modeling stage. These outputs are then summarized locally and aggregated securely through HE, as detailed next.

**Privacy protection in FedIT-CS-M**. We first describe the HE procedure for FedIT-CS-M, which performs discrete client-subset selection. For a candidate subset $\mathcal{I}$, each client $k$ locally computes the summary statistics required for the aggregated correlation:

$$e_x^{(k)} = \sum_{i=1}^{n_k} r_{x;i}^{(k)}, \ e_y^{(k)} = \sum_{i=1}^{n_k} r_{y;i}^{(k)}, \ m_{xy}^{(k)} = \sum_{i=1}^{n_k} r_{x;i}^{(k)} r_{y;i}^{(k)}, \ m_{xx}^{(k)} = \sum_{i=1}^{n_k} r_{x;i}^{(k)} r_{x;i}^{(k)}, \ m_{yy}^{(k)} = \sum_{i=1}^{n_k} r_{y;i}^{(k)} r_{y;i}^{(k)}. \tag{19}$$

Instead of transmitting these quantities in plaintext, client $k$ sends only their encrypted versions:

$$Enc(n_k), \ Enc(e_x^{(k)}), \ Enc(e_y^{(k)}), \ Enc(m_{xx}^{(k)}), \ Enc(m_{xy}^{(k)}), \ Enc(m_{yy}^{(k)}). \tag{20}$$

Because all ciphertexts are generated under the same public key, the server can aggregate them directly in the encrypted domain. For subset $\mathcal{I}$, it obtains

$$Enc(n_\mathcal{I}) = \sum_{k \in \mathcal{I}} Enc(n_k), \ \ Enc(e_x^\mathcal{I}) = \sum_{k \in \mathcal{I}} Enc(e_x^{(k)}), \ \ Enc(e_y^\mathcal{I}) = \sum_{k \in \mathcal{I}} Enc(e_y^{(k)}),$$
$$Enc(m_{xy}^\mathcal{I}) = \sum_{k \in \mathcal{I}} Enc(m_{xy}^{(k)}), \ \ Enc(m_{xx}^\mathcal{I}) = \sum_{k \in \mathcal{I}} Enc(m_{xx}^{(k)}), \ \ Enc(m_{yy}^\mathcal{I}) = \sum_{k \in \mathcal{I}} Enc(m_{yy}^{(k)}).$$

Here, all summations involving encrypted quantities are understood as operations in the ciphertext domain rather than ordinary plaintext addition. We use the same summation notation for brevity, with the intended operation determined by whether its arguments are encrypted or plaintext. The server then transfers the aggregated ciphertexts to a designated decrypting client for decryption:

$$n_\mathcal{I} = Dec(Enc(n_\mathcal{I})), \ e_x^\mathcal{I} = Dec(Enc(e_x^\mathcal{I})), \ e_y^\mathcal{I} = Dec(Enc(e_y^\mathcal{I})),$$
$$m_{xy}^\mathcal{I} = Dec(Enc(m_{xy}^\mathcal{I})), \ m_{xx}^\mathcal{I} = Dec(Enc(m_{xx}^\mathcal{I})), \ m_{yy}^\mathcal{I} = Dec(Enc(m_{yy}^\mathcal{I})). \tag{21}$$

Using these quantities, the decrypting party computes

$$\rho_{xy}^\mathcal{I} = \frac{n_\mathcal{I} m_{xy}^\mathcal{I} - e_x^\mathcal{I} e_y^\mathcal{I}}{\sqrt{n_\mathcal{I} m_{xx}^\mathcal{I} - (e_x^\mathcal{I})^2} \sqrt{n_\mathcal{I} m_{yy}^\mathcal{I} - (e_y^\mathcal{I})^2}}, \tag{22}$$

and returns only this final aggregated result to the server.

Throughout the protocol, no client-specific plaintext statistics, such as $e_x^{(k)}$ or $m_{xx}^{(k)}$, are revealed to the server. The decrypting party receives only subset-level aggregates rather than individual client submissions. This avoids direct exposure of client-level statistics while enabling exact computation of the FedIT-CS-M aggregation criterion.

**Privacy protection in FedIT-CS-ML.** We next describe the privacy-preserving procedure for FedIT-CS-ML, which optimizes the client weights through a continuous, gradient-based formulation. We first recall the relevant aggregated quantities used in this optimization:

$$n_{\mathcal{P}} = \sum_{k \in [K]} p_k n_k, \; e_x^{\mathcal{P}} = \sum_{k \in [K]} p_k e_x^{(k)}, \; e_y^{\mathcal{P}} = \sum_{k \in [K]} p_k e_y^{(k)},$$
$$m_{xy}^{\mathcal{P}} = \sum_{k \in [K]} p_k m_{xy}^{(k)}, \; m_{xx}^{\mathcal{P}} = \sum_{k \in [K]} p_k m_{xx}^{(k)}, \; m_{yy}^{\mathcal{P}} = \sum_{k \in [K]} p_k m_{yy}^{(k)}. \tag{23}$$

Further, we denote

$$\mathfrak{N} := n_{\mathcal{P}} m_{xy}^{\mathcal{P}} - e_x^{\mathcal{P}} e_y^{\mathcal{P}}, \; \mathfrak{A} := n_{\mathcal{P}} m_{xx}^{\mathcal{P}} - (e_x^{\mathcal{P}})^2, \; \mathfrak{B} := n_{\mathcal{P}} m_{yy}^{\mathcal{P}} - (e_y^{\mathcal{P}})^2, \tag{24}$$

and thus the optimizing criterion is given by

$$\rho_{xy}^{\mathcal{P}} = \frac{n_{\mathcal{P}} m_{xy}^{\mathcal{P}} - e_x^{\mathcal{P}} e_y^{\mathcal{P}}}{\sqrt{n_{\mathcal{P}} m_{xx}^{\mathcal{P}} - (e_x^{\mathcal{P}})^2} \sqrt{n_{\mathcal{P}} m_{yy}^{\mathcal{P}} - (e_y^{\mathcal{P}})^2}} =: \frac{\mathfrak{N}}{\sqrt{\mathfrak{A}\mathfrak{B}}}. \tag{25}$$

Then the gradient of $\rho_{xy}^{\mathcal{P}} = \mathfrak{N}/\sqrt{\mathfrak{A}\mathfrak{B}}$ with respect to any $p_k$, $k \in [K]$ is given by

$$\frac{\partial \rho_{xy}^{\mathcal{P}}}{\partial p_k} = \frac{1}{\sqrt{\mathfrak{A}\mathfrak{B}}} \Big( d\mathfrak{N}_k - \frac{\mathfrak{N}}{2} \Big( \frac{d\mathfrak{A}_k}{\mathfrak{A}} + \frac{d\mathfrak{B}_k}{\mathfrak{B}} \Big) \Big), \tag{26}$$

where the terms

$$d\mathfrak{N}_k := \frac{\partial \mathfrak{N}}{\partial p_k} = n_k m_{xy}^{\mathcal{P}} + n_{\mathcal{P}} m_{xy}^{(k)} - e_x^{(k)} e_y^{\mathcal{P}} - e_x^{\mathcal{P}} e_y^{(k)},$$
$$d\mathfrak{A}_k := \frac{\partial \mathfrak{A}}{\partial p_k} = n_k m_{xx}^{\mathcal{P}} + n_{\mathcal{P}} m_{xx}^{(k)} - 2 e_x^{\mathcal{P}} e_x^{(k)}, \; d\mathfrak{B}_k := \frac{\partial \mathfrak{B}}{\partial p_k} = n_k m_{yy}^{\mathcal{P}} + n_{\mathcal{P}} m_{yy}^{(k)} - 2 e_y^{\mathcal{P}} e_y^{(k)}. \tag{27}$$

Below, we detail the HE-protected optimization procedure. In each communication round, client $k$ encrypts and sends its weighted local statistics to the server:

$$Enc(p_k n_k), \; Enc(p_k e_x^{(k)}), \; Enc(p_k e_y^{(k)}), \; Enc(p_k m_{xx}^{(k)}), \; Enc(p_k m_{xy}^{(k)}), \; Enc(p_k m_{yy}^{(k)}). \tag{28}$$

The server then homomorphically aggregates them and returns the encrypted global summaries to the decrypting party:

$$Enc(n_{\mathcal{P}}) = \sum_{k \in [K]} Enc(p_k n_k), \qquad Enc(e_x^{\mathcal{P}}) = \sum_{k \in [K]} Enc(p_k e_x^{(k)}),$$
$$Enc(e_y^{\mathcal{P}}) = \sum_{k \in [K]} Enc(p_k e_y^{(k)}), \qquad Enc(m_{xy}^{\mathcal{P}}) = \sum_{k \in [K]} Enc(p_k m_{xy}^{(k)}),$$
$$Enc(m_{xx}^{\mathcal{P}}) = \sum_{k \in [K]} Enc(p_k m_{xx}^{(k)}), \; Enc(m_{yy}^{\mathcal{P}}) = \sum_{k \in [K]} Enc(p_k m_{yy}^{(k)}). \tag{29}$$

The decrypting party then recovers the aggregated plaintext quantities

$$n_{\mathcal{P}} = Dec(Enc(n_{\mathcal{P}})), \; e_x^{\mathcal{P}} = Dec(Enc(e_x^{\mathcal{P}})), \; e_y^{\mathcal{P}} = Dec(Enc(e_y^{\mathcal{P}})),$$
$$m_{xy}^{\mathcal{P}} = Dec(Enc(m_{xy}^{\mathcal{P}})), \; m_{xx}^{\mathcal{P}} = Dec(Enc(m_{xx}^{\mathcal{P}})), \; m_{yy}^{\mathcal{P}} = Dec(Enc(m_{yy}^{\mathcal{P}})). \tag{30}$$

These aggregated quantities are then made available to the clients for local updates. Together with its own local statistics, client $k$ computes $\partial \rho_{xy}^{\mathcal{P}}/\partial p_k$ by substituting the corresponding values into Eq. (24) and Eq. (27). Throughout this procedure, no client-specific plaintext statistics, such as $e_x^{(k)}$ or $m_{xx}^{(k)}$, are revealed to the server. As a result, HE enables gradient-based optimization while protecting client-level statistics during aggregation.

**Resource consumption of HE.** Homomorphic Encryption is applied only to a small set of low-dimensional aggregated statistics, which substantially reduces computational and communication overhead, especially compared with methods such as FUIT that require transmitting full matrices across clients. We further report the practical cost of the HE component. Using the TenSEAL library, each encryption–decryption cycle takes approximately 4 ms per client, with a communication cost of about 80 KB. The memory footprint is negligible, as measured using `psutil`. Overall, this lightweight design introduces only modest cryptographic overhead, making HE practical for low-latency federated settings.

# E. Preliminaries and Auxiliary Lemmas

In this section, we present the notation and assumptions used in our theoretical analysis, and establish several auxiliary lemmas. These intermediate results provide the foundation for proving the main theorems in the subsequent sections.

## E.1. Assumptions

We begin by stating the assumptions required for our analysis. The first two assumptions are specific to the federated independence testing (FedIT) setting:

**Assumption E.1** (Consistent dependence assumption). The dependence relationship between $X_k$ and $Y_k$ is consistent across clients. Specifically, either $X_k \perp\!\!\!\perp Y_k$ holds for all $k \in [K]$, or $X_k \not\perp\!\!\!\perp Y_k$ holds for all $k \in [K]$.

**Assumption E.2** (Heterogeneous clients assumption). The dependence mechanism (e.g., dependence strength or functional relationship), the marginal distributions $\mathbb{P}_{X_k}, \mathbb{P}_{Y_k}$ and the joint distributions $\mathbb{P}_{X_k, Y_k}$ may vary across clients.

Together, these assumptions define a realistic yet challenging regime: the global dependence status is shared, while local distributions remain heterogeneous. Our goal is to design a test that aggregates cross-client evidence to infer this common dependence status under client heterogeneity.

For the intra-client component, we further impose a mild restriction on the function class in the Hirschfeld–Gebelein–Rényi (HGR) maximum correlation coefficient (Gebelein, 1941), $\mathrm{hgr}(X, Y) = \sup_{f,g} \rho(f(X), g(Y))$, where $\rho$ denotes Pearson's correlation. This restriction provides the functional setting needed for the subsequent approximation analysis.

**Assumption E.3** (Function class assumption). For each client $k \in [K]$, we assume that the optimal transformations $f_k$ and $g_k$ in $\arg\max_{f_k, g_k} \rho(f_k(X_k), g_k(Y_k))$ belong to a reproducing kernel Hilbert space (RKHS) $\mathcal{H}$ associated with a shift-invariant, positive semi-definite, and bounded kernel $k(x, x') = \langle \phi(x), \phi(x') \rangle_{\mathcal{F}} \leq C$.

This RKHS is contained in the broader function class $\mathcal{F}$ considered in Theorem 4.4, ensuring that the random-projection approximation result applies. The assumption also aligns with our practical use of random Fourier features, which provide a tractable approximation while retaining sufficient flexibility to capture nonlinear dependencies.

## E.2. Copula Properties: Marginal Uniformity and Convergence

For completeness, we restate two basic properties of copulas that are repeatedly used in the subsequent analysis. We first recall the marginal uniformity induced by the probability integral transform, and then summarize the finite-sample convergence guarantee for the empirical copula transformation.

**Theorem E.4** (Probability integral transform (Nelsen, 2006)). *For a univariate random variable $X$ with cdf $F$, $U := F(X)$ is uniformly distributed on $[0, 1]$.*

The above result directly implies that the margins of any copula are uniformly distributed on $[0, 1]$, which forms the basis for the copula representation of dependence. Beyond this marginal property, it is also important to understand how well the empirical copula estimates converge to their population counterpart, since our method relies on finite-sample approximations.

**Theorem E.5** (Convergence of the empirical copula (Póczos et al., 2012)). *Let $\boldsymbol{F}$ denote the copula transformation, and let $\boldsymbol{F}_n$ be its empirical counterpart. Then,*

$$\mathbb{P}\left[\sup_{\boldsymbol{x} \in \mathbb{R}^d} \|\boldsymbol{F}(\boldsymbol{x}) - \boldsymbol{F}_n(\boldsymbol{x})\|_2 > \epsilon\right] \leq 2d \exp\left(-\frac{2n\epsilon^2}{d}\right). \tag{31}$$

Together, Theorems E.4 and E.5 show that copula transformations yield uniform marginals and admit strong finite-sample concentration for their empirical approximations. These properties will be used in our subsequent theoretical analysis.

## E.3. Random Projection Properties: Convergence

We next recall a key convergence result for random projections, which underpins their use in our framework. Specifically, the following theorem shows that linear models built on random features can approximate the optimal estimator with controlled error, decreasing in both the sample size $n$ and the number of projections $h$ with high probability.

**Theorem E.6** (Approximation with random projections (Rahimi & Recht, 2008)). *Let $p(w)$ be a distribution on $\Omega$ and assume that $\sup_{x,w} |\phi(x;w)| \leq 1$. Define $\mathcal{F} := \{f(x) = \int_\Omega u(w)\phi(x;w)dw \mid |u(w)| \leq Cp(w)\}$. Draw $w_1, ..., w_h$ i.i.d from $p(w)$. Further let $\delta > 0$, and $c$ be some L-Lipschitz loss function, and consider data $\{x_i, o_i\}_{i=1}^n$ drawn i.i.d from some arbitrary $\mathbb{P}_{XO}$. The linear regression coefficient $u_1, ..., u_h$ defining $f_h(x) = \sum_{j=1}^h u_j\phi(x;w_j)$ are estimated by minimizing the empirical risk $c(f_h(x), o)$. The resulting estimator has a distance from the c-optimal estimator in $\mathcal{F}$ bounded by*

$$\mathbb{E}_\mathbb{P}[c(f_h(x), o)] - \min_{f \in \mathcal{F}} \mathbb{E}_\mathbb{P}[c(f(x), o)] \leq \mathcal{O}\left(\left(\frac{1}{\sqrt{n}} + \frac{1}{\sqrt{h}}\right)LC\sqrt{\log\frac{1}{\delta}}\right) \tag{32}$$

*with probability at least $1 - 2\delta$.*

This result establishes that the approximation error decreases as the number of random features $h$ increases and as the sample size $n$ grows. Hence, random projections provide a computationally efficient way to approximate nonlinear function classes while retaining statistical guarantees. Combined with the empirical copula convergence and the finite-sample convergence of the leading CCA correlation (Hardoon & Shawe-Taylor, 2009), the function class assumption E.3 implies that the correlation between the resulting intra-client projected outputs approaches the population HGR coefficient at rate $\mathcal{O}(1/\sqrt{n} + 1/\sqrt{h})$.

### E.4. Procedure and Properties of FedIT-CS Framework

In this section, we summarize the FedIT-CS procedure and introduce the notation needed for the subsequent analysis. For client $k \in [K]$, let $(X_k, Y_k)$ denote its input variables. The within-client computation proceeds in four steps:

1. **First copula transformation:** map $(X_k, Y_k)$ to $(Q_{X;k}, Q_{Y;k})$ through the client-specific marginal cdfs $F_X^{(k)}$ and $F_Y^{(k)}$. This step removes marginal distributional effects within client $k$ while preserving the underlying dependence structure. By Theorem E.4, $Q_{X;k}$ and $Q_{Y;k}$ have uniform marginals on $[0, 1]$.

2. **Random projection:** embed $(Q_{X;k}, Q_{Y;k})$ into feature-space variables $(\Phi_{X;k}, \Phi_{Y;k})$ using random parameters $(w_X^{(k)}, b_X^{(k)})$ and $(w_Y^{(k)}, b_Y^{(k)})$. This nonlinear feature mapping provides a tractable approximation to rich transformation classes capable of capturing general dependence patterns.

3. **CCA projection:** identify the canonical directions $u^{(k)}$ and $v^{(k)}$ that maximize the correlation between the projected feature representations. We then project $(\Phi_{X;k}, \Phi_{Y;k})$ onto the one-dimensional outputs $(\Psi_{X;k}, \Psi_{Y;k})$, thereby concentrating the dominant intra-client dependence signal into a low-dimensional form.

4. **Second copula transformation:** align the marginals of $(\Psi_{X;k}, \Psi_{Y;k})$ using their cdfs $(F_{\Psi_X}^{(k)}, F_{\Psi_Y}^{(k)})$, yielding the final copula outputs $(R_{X;k}, R_{Y;k})$. This additional alignment facilitates subsequent cross-client aggregation by removing client-specific marginal effects. Again, Theorem E.4 implies that $R_{X;k}$ and $R_{Y;k}$ have uniform marginals on $[0, 1]$.

For clarity, we first set aside the aggregation procedure and focus on the within-client correlation $\rho(R_{X;k}, R_{Y;k})$. This quantity is closely connected to the Hirschfeld–Gebelein–Rényi (HGR) maximum correlation coefficient (Gebelein, 1941), $\text{hgr}(X, Y) = \sup_{f,g} \rho(f(X), g(Y))$, where the supremum is taken over all Borel-measurable functions $f$ and $g$ with finite variance, and $\rho$ denotes Pearson's correlation. In FedIT-CS, the client-side random projection and CCA steps provide a tractable approximation to this maximal dependence characterization. Under Assumption E.3, the optimal transformations lie in a reproducing kernel Hilbert space (RKHS), making the above approximation theoretically well founded.

A classical result is that $\text{hgr}(X, Y)$ satisfies several desirable properties (Rényi, 1959):

1. $\text{hgr}(X, Y)$ is defined for any pair of non-constant random variables $X$ and $Y$.
2. $\text{hgr}(X, Y) = \text{hgr}(Y, X)$.
3. $0 \leq \text{hgr}(X, Y) \leq 1$.
4. $\text{hgr}(X, Y) = 0$ iff $X$ and $Y$ are statistically independent.
5. For bijective Borel-measurable functions $f, g : \mathbb{R} \to \mathbb{R}$, $\text{hgr}(X, Y) = \text{hgr}(f(X), g(Y))$.
6. $\text{hgr}(X, Y) = 1$ if $Y = f(X)$ or $X = g(Y)$ for some Borel-measurable function.
7. If $(X, Y) \sim \mathcal{N}(\mu, \Sigma)$, then $\text{hgr}(X, Y) = |\rho(X, Y)|$.

The invariance of HGR under bijective marginal transformations ensures that copula transformations preserve maximal dependence strength. Together with the random-projection and CCA approximations above, $\rho(R_{X;k}, R_{Y;k})$ retains the key dependence-discriminating behavior of HGR. We summarize this in the following lemma.

**Lemma E.7** (Intra-client dependence characterization). *Under Assumptions E.1, E.2, and E.3, for each client $k \in [K]$, the dependence coefficient $\rho(R_{X;k}, R_{Y;k})$ correctly distinguishes independence from dependence. Specifically, under $\mathcal{H}_0$, we have $\rho(R_{X;k}, R_{Y;k}) = 0$, whereas under $\mathcal{H}_1$, we have $\rho(R_{X;k}, R_{Y;k}) > 0$.*

*Proof.* This follows from the dependence-discriminating property of hgr together with the function class restriction in Assumption E.3. $\qquad\square$

Moreover, the output distribution under $\mathcal{H}_0$ can be characterized more explicitly, as stated below.

**Lemma E.8** (Output copula under $\mathcal{H}_0$). *Under Assumptions E.1, E.2, and E.3, for each client $k \in [K]$, $R_{X;k}, R_{Y;k} \sim \text{Uniform}([0,1] \times [0,1])$ under $\mathcal{H}_0$.*

*Proof.* By Lemma E.7, under $\mathcal{H}_0$, $R_{X;k}$ and $R_{Y;k}$ are independent. In addition, the copula transformation ensures that each marginal is uniformly distributed on $[0,1]$. Combining independence with marginal uniformity yields $(R_{X;k}, R_{Y;k}) \sim \text{Uniform}([0,1] \times [0,1])$. $\qquad\square$

Above, we focused on the idealized theoretical setting, where estimation and approximation errors are ignored. We now turn to the finite-sample case and describe how the within-client computation is implemented in practice. For clarity, we restate the full procedure together with the notation used at each step:

1. **First copula transformation:** given the input samples $\{(x_i^{(k)}, y_i^{(k)})\}_{i=1}^{n_k}$ from $(X_k, Y_k)$, compute the empirical copula samples $\{(q_{x;i}^{(k)}, q_{y;i}^{(k)})\}_{i=1}^{n_k}$ using the empirical marginal cdfs $F_{x;n_k}^{(k)}$ and $F_{y;n_k}^{(k)}$.

2. **Random projection:** map $\{(q_{x;i}^{(k)}, q_{y;i}^{(k)})\}_{i=1}^{n_k}$ to random feature representations $\{(\phi_{x;i}^{(k)}, \phi_{y;i}^{(k)})\}_{i=1}^{n_k}$ using sampled parameters $(w_{x;j}^{(k)}, b_{x;j}^{(k)})$ and $(w_{y;j}^{(k)}, b_{y;j}^{(k)})$ for $j \in [h]$, where $h$ denotes the number of random projections.

3. **CCA projection:** compute the canonical directions $u_h^{(k)}$ and $v_h^{(k)}$, and then project the random features to obtain the one-dimensional outputs $\{(\psi_{x;i}^{(k)}, \psi_{y;i}^{(k)})\}_{i=1}^{n_k}$.

4. **Second copula transformation:** apply the empirical cdfs $F_{\psi_x;n_k}^{(k)}$ and $F_{\psi_y;n_k}^{(k)}$ to $\{(\psi_{x;i}^{(k)}, \psi_{y;i}^{(k)})\}_{i=1}^{n_k}$, yielding the final copula samples $\{(r_{x;i}^{(k)}, r_{y;i}^{(k)})\}_{i=1}^{n_k}$, whose marginals approximate uniform distributions on $[0,1]$.

The same procedure applies to the permutation samples. Following the notation in the main paper, let $\sigma_t$ denote the $t$-th permutation for $t \in [B]$. For client $k$, define

$$\boldsymbol{x}^{(k)} := (x_1^{(k)}, x_2^{(k)}, \ldots, x_{n_k}^{(k)}), \quad \boldsymbol{y}^{(k)} := (y_1^{(k)}, y_2^{(k)}, \ldots, y_{n_k}^{(k)}). \tag{33}$$

Then $(\sigma_t \boldsymbol{x}^{(k)}, \boldsymbol{y}^{(k)})$ denotes the corresponding permuted sample pair. Since permutation only reorders observations without changing their empirical distribution, we obtain the following result.

**Lemma E.9** (Empirical cdf under permutation). *For any $\sigma_t$, $t \in [B]$, the permuted sample $\sigma_t \boldsymbol{x}^{(k)}$ has the same empirical cumulative distribution function $F_{x;n_k}^{(k)}$ as $\boldsymbol{x}^{(k)}$.*

*Proof.* The empirical cumulative distribution function depends only on the multiset of observed values, not on their ordering. Since $\sigma_t$ merely permutes the entries of $\boldsymbol{x}^{(k)}$, its empirical cdf remains unchanged. $\qquad\square$

This property ensures that permutation does not affect the empirical marginal distributions. We next use the permutation invariance under $\mathcal{H}_0$ to characterize the corresponding exchangeability property.

**Lemma E.10** (Exchangeability). *Let $\overset{d}{=}$ denote equality in distribution. Under $\mathcal{H}_0$, for each client $k$, the sequence*

$$(\boldsymbol{x}^{(k)}, \boldsymbol{y}^{(k)}), \ (\sigma_1 \boldsymbol{x}^{(k)}, \boldsymbol{y}^{(k)}), \ \ldots, \ (\sigma_B \boldsymbol{x}^{(k)}, \boldsymbol{y}^{(k)}) \tag{34}$$

*is jointly exchangeable, meaning that its distribution is invariant to reordering the original and permuted sample pairs.*

*Proof.* Under $\mathcal{H}_0$, $X_k$ and $Y_k$ are independent. Therefore, for every permutation $\sigma_t$, $t \in [B]$, reordering $\boldsymbol{x}^{(k)}$ does not alter its joint distribution with $\boldsymbol{y}^{(k)}$. Hence,

$$(\sigma_t \boldsymbol{x}^{(k)}, \boldsymbol{y}^{(k)}) \stackrel{d}{=} (\boldsymbol{x}^{(k)}, \boldsymbol{y}^{(k)}). \tag{35}$$

Since all original and permuted sample pairs share the same distribution, and their joint law is invariant to reordering their indices, the sequence is exchangeable by definition. $\square$

Together, Lemmas E.9 and E.10 show that exchangeability is preserved through the within-client pipeline.

**Lemma E.11** (Exchangeability of client-level output copulas)**.** *Fix a client $k$, and let $\mathcal{T}$ denote the four-step within-client pipeline consisting of (i) empirical copula transformation, (ii) random projection, (iii) CCA projection, and (iv) a second empirical copula transformation. Under $\mathcal{H}_0$ (i.e., $X_k \perp\!\!\!\perp Y_k$), the sequence*

$$\mathcal{T}(\boldsymbol{x}^{(k)}, \boldsymbol{y}^{(k)}), \ \mathcal{T}(\sigma_1 \boldsymbol{x}^{(k)}, \boldsymbol{y}^{(k)}), \dots, \mathcal{T}(\sigma_B \boldsymbol{x}^{(k)}, \boldsymbol{y}^{(k)}) \tag{36}$$

*is exchangeable.*

*Proof.* We show that exchangeability is preserved at each stage of the mapping $\mathcal{T}$. To streamline the argument, additional notation will be introduced as needed within the proof.

By Lemma E.10, under $\mathcal{H}_0$, the input sequence $(\boldsymbol{x}^{(k)}, \boldsymbol{y}^{(k)})$, $(\sigma_1 \boldsymbol{x}^{(k)}, \boldsymbol{y}^{(k)}), \dots, (\sigma_B \boldsymbol{x}^{(k)}, \boldsymbol{y}^{(k)})$ is exchangeable.

*Step (i): First copula transformation.* By Lemma E.9, permuting $\boldsymbol{x}^{(k)}$ does not alter its empirical cdf $F_{x;n_k}^{(k)}$; the empirical cdf of $\boldsymbol{y}^{(k)}$ is unchanged as well. Hence, the same empirical copula transformation is applied to each element of the exchangeable input sequence, which preserves exchangeability of the resulting copula samples.

*Step (ii): Random projection.* Condition on the sampled random-feature parameters used in the projection step. Given these parameters, the random feature map is a deterministic measurable transformation applied identically to the original and permuted copula samples. Therefore, exchangeability of the copula samples implies exchangeability of the projected feature pairs. Since the argument holds conditionally on the sampled parameters, it also holds unconditionally.

*Step (iii): CCA projection.* We view the CCA step as a deterministic measurable operator. Let $(\Phi_{X;t}^{(k)}, \Phi_{Y;t}^{(k)})$, $t = 0, 1, \dots, B$, denote the projected feature matrices obtained from the original sample and the permuted samples. Define the local CCA operator $\mathcal{C}$ by $(u_t^{(k)}, v_t^{(k)}, \rho_t^{(k)}) = \mathcal{C}(\Phi_{X;t}^{(k)}, \Phi_{Y;t}^{(k)})$. Concretely, $\mathcal{C}$ computes the empirical covariance matrices $C_{xx;t}^{(k)}$, $C_{yy;t}^{(k)}$, and $C_{xy;t}^{(k)}$, forms $K_t^{(k)} = (C_{xx;t}^{(k)})^{-1/2} C_{xy;t}^{(k)} (C_{yy;t}^{(k)})^{-1/2}$, and extracts the leading singular value $\rho_t^{(k)} = \sigma_{\max}(K_t^{(k)})$ together with the associated canonical directions $u_t^{(k)}$ and $v_t^{(k)}$. To make $\mathcal{C}$ single-valued, we adopt a fixed deterministic convention for resolving the usual SVD ambiguities, such as sign indeterminacy and possible ties among singular values. For example, one may impose a fixed orientation rule on the canonical vectors; any deterministic tie-breaking rule suffices for the argument. Under such a convention, $\mathcal{C}$ is a measurable functional of the projected feature matrices.

Since the sequence $(\Phi_{X;0}^{(k)}, \Phi_{Y;0}^{(k)}), \dots, (\Phi_{X;B}^{(k)}, \Phi_{Y;B}^{(k)})$ is exchangeable after Step (ii), applying the same measurable map $\mathcal{C}$ to each element preserves exchangeability. Consequently, the CCA outputs $(u_t^{(k)}, v_t^{(k)}, \rho_t^{(k)})$, $t = 0, 1, \dots, B$, are exchangeable. The resulting one-dimensional projected variables $\Psi_{X;t}^{(k)} = (u_t^{(k)})^\top \Phi_{X;t}^{(k)}$ and $\Psi_{Y;t}^{(k)} = (v_t^{(k)})^\top \Phi_{Y;t}^{(k)}$ are deterministic functions of these CCA outputs and therefore remain exchangeable.

*Step (iv): Second copula transformation.* The second empirical copula transformation is applied separately and identically to each projected pair $(\Psi_{X;t}^{(k)}, \Psi_{Y;t}^{(k)})$. Since this step is again a deterministic measurable mapping, exchangeability of the CCA-projected outputs implies exchangeability of the final copula outputs.

Combining all steps, the sequence of outputs across permutations is exchangeable. $\square$

*Remark.* The exchangeability argument in the CCA step only requires the CCA solver to be a deterministic measurable functional of its input feature matrices. The fixed tie-breaking and orientation conventions introduced above are imposed merely to make this property explicit for analysis. In practice, the CCA projection is followed by a copula transformation, so the final output statistics depend only on empirical ranks. This makes the procedure invariant to monotonic rescaling and robust to symmetric sign flips, further reducing the practical relevance of orientation choices.

As a consequence, the client-level output copula sequence remains exchangeable under the null hypothesis $\mathcal{H}_0$, which will be used to establish finite-sample Type I error control for our test .

# F. Proof of Theorem 5.1

Next, we study the theoretical properties of the aggregated statistic. The client-level results and notation have been introduced in Appendix E; here, we focus on the aggregation step. Let $\mathcal{A}$ denote the aggregation algorithm that determines the weights $p_k$, $k \in [K]$. When $p_k \in \{0, 1\}$, this corresponds to FedIT-CS-M, whereas allowing $p_k \in [0, 1]$ corresponds to FedIT-CS-ML. We consider the oracle setting in which $\mathcal{A}$ attains the theoretically optimal weights. In the asymptotic regime where both the sample size and the number of random features are sufficiently large, let $\rho_{\mathcal{A}}$ denote the resulting aggregated coefficient. The following theorem establishes its theoretical soundness.

**Theorem F.1** (Soundness of aggregated statistic). *Let $\rho_{\mathcal{A}}$ denote the aggregated coefficient obtained by the aggregation algorithm $\mathcal{A}$. Assume $\mathcal{A}$ is theoretically optimal. Then, under the null hypothesis $\mathcal{H}_0$, we have $\rho_{\mathcal{A}} = 0$, whereas under the alternative hypothesis $\mathcal{H}_1$, we have $\rho_{\mathcal{A}} > 0$.*

*Proof.* We first consider $\mathcal{H}_0$. By Lemmas E.7 and E.8, for every client $k \in [K]$, we have $\rho(R_{X;k}, R_{Y;k}) = 0$, and the output pair $(R_{X;k}, R_{Y;k})$ follows the uniform distribution on $[0, 1]^2$. Hence, all clients share identical marginal moments and zero cross-covariance. Therefore, any weighted aggregation preserves zero covariance between the aggregated outputs, implying $\rho_{\mathcal{A}} = 0$. Next, we consider the case under $\mathcal{H}_1$. In this setting, by Lemma E.7, each client exhibits a strictly positive dependence coefficient $\rho(R_{X;k}, R_{Y;k}) > 0$, reflecting the underlying dependence between $X_k$ and $Y_k$. Since the aggregation algorithm $\mathcal{A}$ is assumed to be theoretically optimal, it assigns weights $\{p_k\}_{k=1}^K$ in a way that maximizes the global statistic, thus $\rho_{\mathcal{A}} \geq \max_k \rho(R_{X;k}, R_{Y;k}) > 0$, which completes the whole proof. $\square$

# G. Proof of Theorem 5.2

In this section, we prove the finite-sample Type I error bound of our proposed test. The client-level exchangeability results and the associated notation have been introduced in Appendix E. We now lift these properties to the aggregated statistic.

Let the optimized aggregation weights be $\boldsymbol{p}^* = (p_1^*, p_2^*, \ldots, p_K^*)$. These weights are learned on the training split and are therefore fixed with respect to the permutation samples used for testing. During the testing stage, each client $k$ produces the empirical copula outputs $(\boldsymbol{r}_x^{(k)}, \boldsymbol{r}_y^{(k)})$ from its local sample pair $(\boldsymbol{x}^{(k)}, \boldsymbol{y}^{(k)})$. For the $t$-th permuted input $(\sigma_t \boldsymbol{x}^{(k)}, \boldsymbol{y}^{(k)})$, we denote the corresponding output copula vectors by

$$\left(\boldsymbol{r}_x^{(k,\sigma_t)}, \boldsymbol{r}_y^{(k,\sigma_t)}\right) := \mathcal{T}\left(\sigma_t \boldsymbol{x}^{(k)}, \boldsymbol{y}^{(k)}\right), \ t \in [B], \tag{37}$$

where $\mathcal{T}$ denotes the full within-client transformation pipeline defined in Appendix E. Based on these definitions, we establish the exchangeability of the aggregated statistic.

**Proposition G.1** (Exchangeability of the aggregated statistic). *Let the optimized aggregation weights be $\boldsymbol{p}^* = (p_1^*, \ldots, p_K^*)$, and assume that they are fixed with respect to the permutation samples used for testing. For each client $k$, let $(\boldsymbol{r}_x^{(k)}, \boldsymbol{r}_y^{(k)})$ denote the empirical output copula vectors computed from the observed pair $(\boldsymbol{x}^{(k)}, \boldsymbol{y}^{(k)})$. For the $t$-th permuted input $(\sigma_t \boldsymbol{x}^{(k)}, \boldsymbol{y}^{(k)})$, let $(\boldsymbol{r}_x^{(k,\sigma_t)}, \boldsymbol{r}_y^{(k,\sigma_t)})$ denote the corresponding output copula vectors obtained after applying the same within-client transformation pipeline. For the observed outputs, define $e_x^{\mathcal{P}} = \sum_{k=1}^K p_k^* e_x^{(k)}$, $e_y^{\mathcal{P}} = \sum_{k=1}^K p_k^* e_y^{(k)}$, $m_{xy}^{\mathcal{P}} = \sum_{k=1}^K p_k^* m_{xy}^{(k)}$, $m_{xx}^{\mathcal{P}} = \sum_{k=1}^K p_k^* m_{xx}^{(k)}$, $m_{yy}^{\mathcal{P}} = \sum_{k=1}^K p_k^* m_{yy}^{(k)}$, and $n_{\mathcal{P}} = \sum_{k=1}^K p_k^* n_k$. The aggregated Pearson-type statistic $\rho_{xy}^{\mathcal{P}}$ is then computed as in Eq. (16). For notational convenience, write $\rho_{xy}^{\mathcal{P},\sigma_0} := \rho_{xy}^{\mathcal{P}}$ for the observed statistic, and let $\rho_{xy}^{\mathcal{P},\sigma_t}$, $t \in [B]$, denote the corresponding aggregated statistic computed from $\{(\boldsymbol{r}_x^{(k,\sigma_t)}, \boldsymbol{r}_y^{(k,\sigma_t)})\}_{k=1}^K$. Under $\mathcal{H}_0$, the sequence $\rho_{xy}^{\mathcal{P},\sigma_0}, \rho_{xy}^{\mathcal{P},\sigma_1}, \ldots, \rho_{xy}^{\mathcal{P},\sigma_B}$ is exchangeable.*

*Proof.* We proceed in four steps.

*Step 1: Exchangeability of client-level output copulas.* By Lemma E.11, under $\mathcal{H}_0$, for each client $k$, the sequence $(\boldsymbol{r}_x^{(k)}, \boldsymbol{r}_y^{(k)})$, $(\boldsymbol{r}_x^{(k,\sigma_1)}, \boldsymbol{r}_y^{(k,\sigma_1)})$, $\ldots$, $(\boldsymbol{r}_x^{(k,\sigma_B)}, \boldsymbol{r}_y^{(k,\sigma_B)})$ is exchangeable. Since clients are mutually independent and the permutation procedure is applied symmetrically across $t = 0, 1, \ldots, B$, the full collection indexed by $t$ is jointly exchangeable.

*Step 2: Exchangeability of client-level summary statistics.* For any pair of output vectors $(\boldsymbol{u}, \boldsymbol{v})$, define $e_u = \sum_i u_i$, $e_v = \sum_i v_i$, $m_{uu} = \sum_i u_i^2$, $m_{vv} = \sum_i v_i^2$, and $m_{uv} = \sum_i u_i v_i$. These are deterministic measurable functions of $(\boldsymbol{u}, \boldsymbol{v})$. Applying them to the exchangeable sequence in Step 1 preserves exchangeability. Hence, for each client $k$, the sequences $\{e_x^{(k,\sigma_t)}\}_{t=0}^B$, $\{e_y^{(k,\sigma_t)}\}_{t=0}^B$, $\{m_{xx}^{(k,\sigma_t)}\}_{t=0}^B$, $\{m_{yy}^{(k,\sigma_t)}\}_{t=0}^B$, and $\{m_{xy}^{(k,\sigma_t)}\}_{t=0}^B$ are exchangeable.

*Step 3: Fixed-weight aggregation preserves exchangeability.* For each permutation index $t$, define the aggregated statistics $e_x^{\mathcal{P},\sigma_t} = \sum_{k=1}^K p_k^* e_x^{(k,\sigma_t)}$, $e_y^{\mathcal{P},\sigma_t} = \sum_{k=1}^K p_k^* e_y^{(k,\sigma_t)}$, $m_{xy}^{\mathcal{P},\sigma_t} = \sum_{k=1}^K p_k^* m_{xy}^{(k,\sigma_t)}$, with $m_{xx}^{\mathcal{P},\sigma_t}$ and $m_{yy}^{\mathcal{P},\sigma_t}$ defined analogously. The effective sample size $n_{\mathcal{P}} = \sum_{k=1}^K p_k^* n_k$ is fixed across permutations. Since the weights $\{p_k^*\}_{k=1}^K$ do not depend on $t$, these aggregated quantities are fixed linear combinations of exchangeable client-level statistics, and therefore remain exchangeable.

*Step 4: Exchangeability of the aggregated correlation statistic.* The mapping from $(e_x^{\mathcal{P}}, e_y^{\mathcal{P}}, m_{xx}^{\mathcal{P}}, m_{yy}^{\mathcal{P}}, m_{xy}^{\mathcal{P}}, n_{\mathcal{P}})$ to $\rho_{xy}^{\mathcal{P}}$ in Eq. (16) is deterministic and measurable whenever the denominator is nonzero. Applying this same map to the exchangeable aggregated summaries preserves exchangeability. Therefore, $\{\rho_{xy}^{\mathcal{P},\sigma_t}\}_{t=0}^B$ is exchangeable under $\mathcal{H}_0$. □

As a direct consequence, the Type I error of our proposed test is provably controlled.

**Theorem G.2** (Type I error bound). *Assume the null hypothesis $\mathcal{H}_0$ is true. For any significance level $\alpha \in (0,1)$, the bound for the Type I error is given by*

$$\mathbb{P}(\text{p-value} \leq \alpha | \mathcal{H}_0) \leq \alpha. \tag{38}$$

*Proof.* For simplicity, we also write $\mathbb{P}(\cdot | \mathcal{H}_0)$ as $\mathbb{P}_{\mathcal{H}_0}$. For any given $\alpha \in (0,1)$, we have

$$\begin{aligned}
\mathbb{P}_{\mathcal{H}_0}(\text{p-value} \leq \alpha) &= \mathbb{P}_{\mathcal{H}_0}\left(\frac{1 + \sum_{t=1}^B \mathbb{I}\{\rho_{xy}^{\mathcal{P},\sigma_t} \geq \rho_{xy}^{\mathcal{P},\sigma_0}\}}{1+B} \leq \alpha\right) \\
&\leq \mathbb{P}_{\mathcal{H}_0}\left(\sum_{t=1}^B \mathbb{I}\{\rho_{xy}^{\mathcal{P},\sigma_t} \geq \rho_{xy}^{\mathcal{P},\sigma_0}\} \leq \lfloor \alpha(1+B) \rfloor\right).
\end{aligned} \tag{39}$$

Since the sequence $\{\rho_{xy}^{\mathcal{P},\sigma_t}\}_{t=0}^B$ is exchangeable, by the property of order statistics, we have

$$\mathbb{P}_{\mathcal{H}_0}\left(\sum_{t=1}^B \mathbb{I}\{\rho_{xy}^{\mathcal{P},\sigma_t} \geq \rho_{xy}^{\mathcal{P},\sigma_0}\} \leq \lfloor \alpha(1+B) \rfloor\right) = \frac{\lfloor \alpha(1+B) \rfloor}{1+B} \leq \alpha, \tag{40}$$

which completes the proof. □

*Remark.* The finite-sample Type I error bound in Theorem G.2 is independent of both the number of clients $K$ and the number of permutations $B$. The role of $B$ is to determine the resolution of the permutation $p$-value, whose minimum attainable value is $1/(B+1)$. Thus, larger $B$ yields a finer approximation of the rejection threshold, whereas smaller $B$ may make the test more conservative; in particular, if $1/(B+1) > \alpha$, rejection cannot occur. Nevertheless, the bound $\mathbb{P}(\text{p-value} \leq \alpha \mid \mathcal{H}_0) \leq \alpha$ remains valid for any $B$. The number of clients $K$ affects the construction of the aggregated statistic and may improve testing power by incorporating more cross-client evidence. However, under $\mathcal{H}_0$, the exchangeability argument used above remains valid regardless of $K$, so the Type I error guarantee is unchanged.

## H. Details of the Experimental Setup

In this section, we provide detailed descriptions of the experimental settings used in the main paper, including dataset specifications and implementation details. These settings are kept consistent across experiments unless stated otherwise, ensuring fair comparisons among all methods.

**Implementation details.** All methods use $h = 10$ random features and significance level $\alpha = 0.05$. For FUIT, we adopt the median bandwidth setting. For methods requiring data splitting, namely FedIT-CS-M and FedIT-CS-ML, the split ratio is fixed at 0.2. For FedIT-CS-M-F and FedIT-CS-ML-F, where the aggregation strategy is trained on additional data not used for testing, the extra data are set by default to have the same size as the testing data. For FedIT-CS-ML and FedIT-CS-ML-F, which involve gradient-based optimization, the number of iterations is set to 100. Unless otherwise specified, the number of permutations is fixed at $B = 100$. All experiments are conducted on the same hardware platform equipped with an 8-core CPU. These implementation choices are used throughout the reported results to maintain a unified evaluation protocol.

## H.1. Details about Synthetic Data Experiments

Below, we provide the setup details of the synthetic datasets, which include three heterogeneous scenarios: covariance heterogeneity, frequency heterogeneity, and functional heterogeneity. These settings are designed to evaluate the capability of our method under different types of distributional shifts. For this part, the number of clients is $K = 3$.

*Covariance.* We consider a heterogeneous scenario where clients follow distinct covariance structures. The data generation process is specified as follows:

- **Client 1:** $X \sim \mathcal{N}(0,1), \quad Y = 0.5X + \epsilon_1, \ \epsilon_1 \sim \mathcal{N}(0,1);$
- **Client 2:** $X \sim \mathcal{N}(0,1), \quad Y = -0.5X + \epsilon_2, \ \epsilon_2 \sim \mathcal{N}(0,1);$
- **Client 3:** $X \sim \mathcal{N}(0,1), \quad Y = 0.02X + \epsilon_3, \ \epsilon_3 \sim \mathcal{N}(0,1).$

The sample sizes for the three clients follow the ratio $n_1 : n_2 : n_3 = 1 : 1 : 2$. We vary the sample size of Client 1 with $n_1 \in \{100, 150, 200, 250, 300, 400\}$, and scale the other clients proportionally.

*Frequency.* We next consider a heterogeneous scenario where clients exhibit distinct frequency parameters. Specifically, we adopt the sinusoid model (Sejdinovic et al., 2013) with density

$$(X, Y) \sim \mathbb{P}_{xy}(x, y) \propto 1 + \sin(\omega x) \sin(\omega y), \quad (x, y) \in [-\pi, \pi] \times [-\pi, \pi], \tag{41}$$

where $\omega$ denotes the frequency. Larger $\omega$ values make the distribution closer to Uniform$([-\pi, \pi]^2)$, thereby increasing the difficulty of detecting dependence for small sample sizes. We assign client-specific frequencies $\omega_1 = 2$, $\omega_2 = 3$, and $\omega_3 = 4$. The sample sizes follow the ratio $n_1 : n_2 : n_3 = 1 : 1 : 1$. We vary the sample size of Client 1 with $n_1 \in \{100, 150, 200, 250, 300, 400\}$, and scale the other clients proportionally.

*Functional.* Finally, we consider a heterogeneous scenario where clients follow distinct functional relationships. The data generation procedure is defined as follows:

- **Client 1:** $X \sim \text{Uniform}(0,1), \quad Y = \sin(X) + \epsilon_1, \ \epsilon_1 \sim \mathcal{N}(0,1);$
- **Client 2:** $X \sim \text{Uniform}(0,1), \quad Y = \cos(X) + \epsilon_2, \ \epsilon_2 \sim \mathcal{N}(0,1);$
- **Client 3:** $X \sim \text{Uniform}(0,1), \quad Y = X^2 + \epsilon_3, \ \epsilon_3 \sim \mathcal{N}(0,1).$

The sample sizes for the three clients follow the ratio $n_1 : n_2 : n_3 = 4 : 2 : 1$. We vary the sample size of Client 1 with $n_1 \in \{400, 600, 800, 1000, 1200, 1400, 1600\}$, and scale the other clients proportionally.

## H.2. Details about Real Data Experiments

**Sachs dataset.** To evaluate the effectiveness of our proposed method in real-world, we employed the well-known Sachs (Sachs et al., 2005) dataset under seven perturbation conditions: (i) anti-CD3 + anti-CD28, (ii) anti-CD3/CD28 + ICAM-2, (iii) anti-CD3/CD28 + U0126, (iv) anti-CD3/CD28 + AKT inhibitor, (v) anti-CD3/CD28 + G06976, (vi) anti-CD3/CD28 + Psitectorigenin, and (vii) anti-CD3/CD28 + LY294002. In our setting, each perturbation condition in the Sachs dataset is regarded as a distinct client. These perturbations cover both general T-cell activation and specific pathway-targeted interventions, thereby enabling a diverse range of causal effects within the signaling network. The causal network is presented in Fig. 5. This network comprises 11 nodes and 18 arcs, and is commonly recognized as a benchmark ground truth. It has been extensively adopted in prior studies on causal discovery (Zhang et al., 2023b). The 11 nodes of this network form 55 distinct node pairs in total. Among these pairs, 18 are independent of each other, whereas the remaining 37 exhibit a dependent relationship. We further visualize the distributions of all 11 variables under each experimental condition, and the results are presented in Fig. 6, where each row corresponds to one perturbation condition and each column corresponds to one variable. *From the marginal distributions, we can observe that the variable distributions change under different perturbation conditions, which aligns with our heterogeneity assumption.*

**Experimental setup.** In our setting, each perturbation condition in the Sachs dataset is treated as a distinct client, yielding a total of seven clients. We compare our proposed methods against FUIT, FedIT-CS-S, FedIT-CS-M, and FedIT-CS-ML. At each iteration, 3 clients are randomly sampled from 7 clients, and we evaluate all 55 node pairs: 18 independent pairs for Type I error assessment and 37 dependent pairs for Type II error assessment. To ensure statistical reliability, this procedure is repeated 50 times with independent client selections, and we report the averaged performance across all trials.

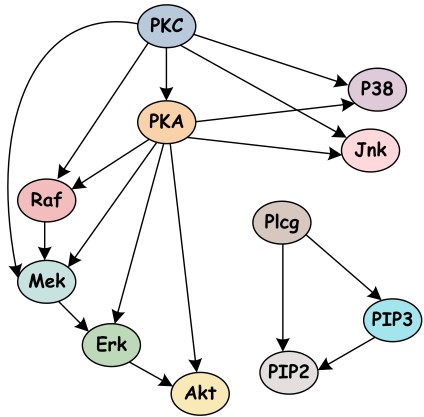

*Figure 5.* The causal graph of Sachs network.

*Figure 6.* Distribution of the 11 variables across seven perturbation conditions in the Sachs dataset.

# I. Additional Experiment Results

In this section, we report additional experiments under broader settings, including alternative noise distributions, heterogeneous functional relationships, and larger client scales. We also provide supplementary comparisons with aggregation strategies, runtime results, real-world validation, and additional baselines.

## I.1. Results with Diverse Distributions

In the synthetic data experiments under the *Covariance* setting, we assume the input noise follows a Gaussian distribution. In the following, we further examine the results under alternative noise distributions. Specifically, we fix the number of clients to $K = 3$, with their sample sizes following the ratio $n_1 : n_2 : n_3 = 1 : 1 : 2$. We vary the sample size of Client 1 with $n_1 \in \{100, 150, 200, 250, 300, 400\}$. We conduct 100 independent trials and report the average results.

For the alternative hypothesis $\mathcal{H}_1$, the data are generated as follows:

- **Client 1:** $X \sim \text{Distribution}(\cdot),\quad Y = 0.5X + \epsilon,\ \epsilon \sim \text{Distribution}(\cdot)$;
- **Client 2:** $X \sim \text{Distribution}(\cdot),\quad Y = -0.5X + \epsilon,\ \epsilon \sim \text{Distribution}(\cdot)$;
- **Client 3:** $X \sim \text{Distribution}(\cdot),\quad Y = 0.02X + \epsilon,\ \epsilon \sim \text{Distribution}(\cdot)$.

For the null hypothesis $\mathcal{H}_0$, the data are generated as follows:

- **Client 1:** $X \sim \text{Distribution}(\cdot),\quad Y \sim \text{Distribution}(\cdot)$;
- **Client 2:** $X \sim \text{Distribution}(\cdot),\quad Y \sim \text{Distribution}(\cdot)$;
- **Client 3:** $X \sim \text{Distribution}(\cdot),\quad Y \sim \text{Distribution}(\cdot)$.

Here, $\text{Distribution}(\cdot)$ is drawn from $\{\text{Laplace}(0, 1), \text{Uniform}(-2, 2)\}$.

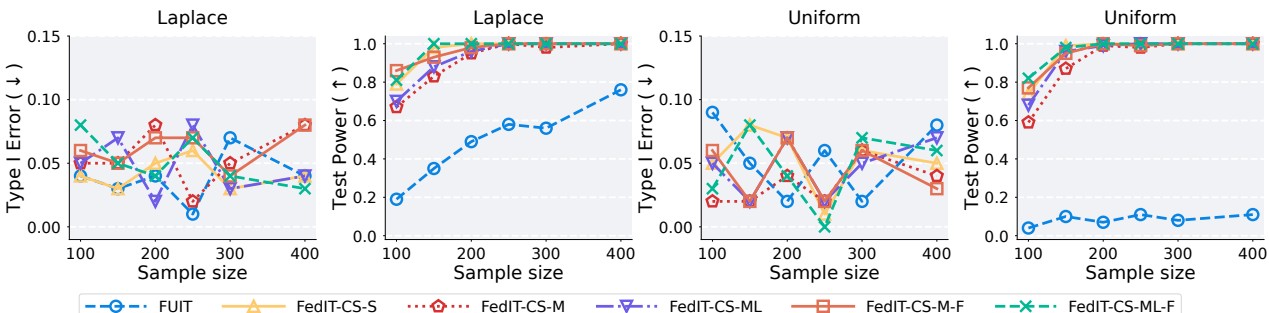

*Figure 7.* Results under Laplace and Uniform noise distributions.

**Performance and analysis.** The results are shown in Fig. 7. All methods control the Type I error rate across the considered distributions and sample sizes. Compared with FUIT, our variants consistently achieve higher power under both Laplace and Uniform noise, supporting the effectiveness of stacking-based aggregation. The gap between the "-F" and non-"-F" variants further indicates the power loss caused by sample splitting. Although FedIT-CS-ML was motivated as a scalable alternative to FedIT-CS-M, its more flexible continuous aggregation can also yield stronger detection power.

## I.2. Results across Functional Relationships and Client Scales

The preceding experiments focus on three clients with relatively simple dependence structures. We now extend the evaluation to more diverse functional relationships and larger client scales, considering both linear and nonlinear settings.

***Linear case.*** The data-generating mechanisms are specified as

$$\begin{aligned}
\mathcal{H}_1 : \quad & X \sim \mathcal{N}(0, 1), \qquad Y = aX + \epsilon_0,\ \epsilon_0 \sim \mathcal{N}(0, 1), \\
\mathcal{H}_0 : \quad & X \sim \mathcal{N}(0, 1), \qquad Y = a\epsilon_1 + \epsilon_2,\ \epsilon_1, \epsilon_2 \sim \mathcal{N}(0, 1),
\end{aligned} \tag{42}$$

where $a \sim \text{Uniform}(-0.5, 0.5)$ is a randomly sampled slope parameter.

***Non-linear case.*** The data-generating mechanisms are specified as

$$\begin{aligned} \mathcal{H}_1: \quad & X \sim \mathcal{N}(0,1), \qquad Y = f(X + \epsilon_0) + \epsilon_1, \ \ \epsilon_0, \epsilon_1 \sim \mathcal{N}(0,1), \\ \mathcal{H}_0: \quad & X = f(\epsilon_2), \qquad Y = f(\epsilon_3) + \epsilon_4, \ \ \epsilon_2, \epsilon_3, \epsilon_4 \sim \mathcal{N}(0,1), \end{aligned} \tag{43}$$

where $f(\cdot)$ is randomly chosen from $\{\sin(\cdot), \cos(\cdot), \tanh(\cdot), \exp(-|\cdot|), (\cdot)^2\}$, and all noise terms are mutually independent. This construction induces heterogeneous nonlinear relationships across clients.

To study the effect of the number of clients, we fix the sample size per client at $n = 200$ and vary $K \in \{2, 3, 4, 5, 6\}$. Each configuration is repeated over 100 independent trials, and the average results are reported in Fig. 8.

We further evaluate large-scale federated settings by varying the number of clients as $K \in \{10, 20, 30, 40, 50\}$, with each client holding 50 samples. The data generation follows the linear case above. Each configuration is repeated over 100 trials, and the results are summarized in Table 3.

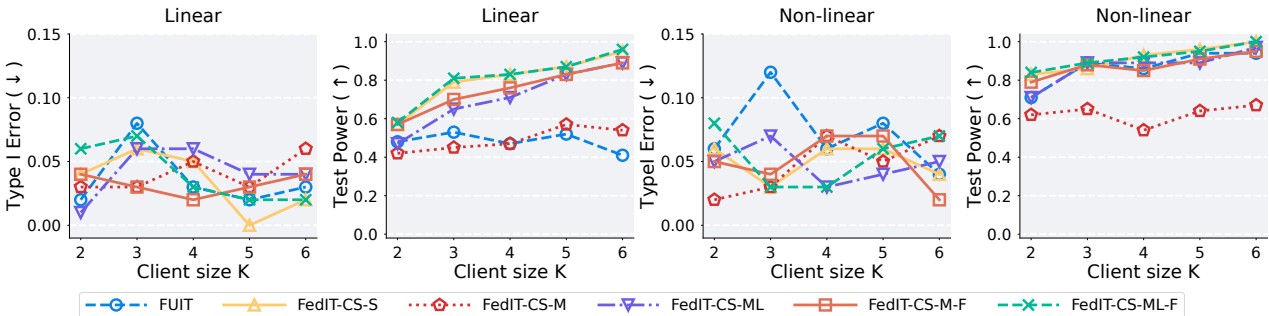

*Figure 8.* Results across heterogeneous functional relationships and client scales.

**Results of Linear case.** In the linear setting, all methods successfully control the Type I error rate. FedIT-CS-ML achieves higher power than FUIT and FedIT-CS-M, highlighting the benefit of its continuous aggregation strategy. FedIT-CS-M is comparable to FUIT for smaller $K$, but becomes more favorable as the number of clients increases. The gap between the "-F" and non-"-F" variants again reflects the power loss caused by sample splitting, especially for FedIT-CS-M, whose subset selection becomes less reliable with limited split data. Moreover, FUIT gains little from increasing $K$, consistent with the dependence dilution induced by naive feature-space aggregation under heterogeneity.

**Results of the Non-linear case.** Except for FUIT under the nonlinear setting with $K = 3$, all methods successfully control the Type I error rate. Regarding detection power, FedIT-CS-S achieves the best performance in this setting, while FedIT-CS-ML and FUIT also perform well. In contrast, FedIT-CS-M shows inferior performance; comparing it with FedIT-CS-M-F suggests that sample splitting leads to a power loss. Since our split ratio is set to $0.2$, the selection component becomes ineffective with the resulting small training sample size (only $100 \times 0.2 = 20$ for each client), which explains the degraded performance. By contrast, FedIT-CS-ML, though based on the same sample size, still achieves strong results.

*Table 3.* Power comparison across $K \in [10, 50]$ clients, where each client has $n_k = 50$ samples and the total sample size is $N = 50K$.

| Clients $K$ | Total $N$ | FUIT | FedIT-CS-S | FedIT-CS-ML |
|---|---|---|---|---|
| 10 | 500 | 0.18 | 0.17 | 0.13 |
| 20 | 1000 | 0.26 | 0.39 | 0.30 |
| 30 | 1500 | 0.30 | 0.61 | 0.51 |
| 40 | 2000 | 0.28 | 0.72 | 0.55 |
| 50 | 2500 | 0.31 | 0.83 | 0.64 |

**Results in larger-scale federated settings.** Our FedIT variants demonstrate strong scalability. In particular, the power of FedIT-CS-S increases monotonically from $0.17$ at $K = 10$ to $0.83$ at $K = 50$. By contrast, FUIT gains limited power as the total sample size grows, remaining around $0.30$. This suggests that FUIT is less effective at exploiting collective information from many clients when local sample sizes are small. These results show that our framework can better leverage increasing client participation to mitigate local data scarcity, making it suitable for larger-scale federated settings.

## I.3. Comparison with More Aggregation Strategies

In Sec. 4.3, we argue that the aggregated correlation coefficient already provides an effective criterion for client selection. Here, we empirically examine this choice by comparing it with a permutation-based alternative, termed FedIT-CS-MB. Specifically, FedIT-CS-MB is derived from FedIT-CS-M by replacing the correlation-based power proxy with a permutation-based estimate: each client transmits $B + 1$ sets of summary statistics to the server, which are then used to estimate $p$-values and approximate testing power. We compare FedIT-CS-MB and FedIT-CS-M in terms of both performance and efficiency. For the error-rate comparison, we follow the covariance setting in Sec. H.1 and fix the baseline sample size at $n_1 = 150$. For the runtime comparison, we adopt the linear setting in Sec. I.2, where the per-client sample size is fixed at $n = 200$ and the number of clients varies as $K \in \{2, 4, 8, 16\}$ to assess computational scalability.

**Performance and analysis.** The results are shown in the left two panels of Fig. 9. The leftmost panel compares Type I and Type II error rates. Both methods successfully control the Type I error rate, while FedIT-CS-M achieves a lower Type II error rate than FedIT-CS-MB. This suggests that the correlation-coefficient-based aggregation criterion already captures sufficient dependence information without relying on permutation-based power modeling. The middle panel reports runtime. Although both methods have theoretical complexity $\mathcal{O}(2^K)$ due to subset enumeration, FedIT-CS-MB incurs an additional multiplicative factor of $B$ from permutation-based score estimation, which substantially worsens scalability. As shown, FedIT-CS-MB handles at most 8 clients within roughly 500 seconds, whereas FedIT-CS-M processes 16 clients in about 20 seconds. Overall, FedIT-CS-M provides a more favorable trade-off, achieving better empirical performance with substantially lower computational cost.

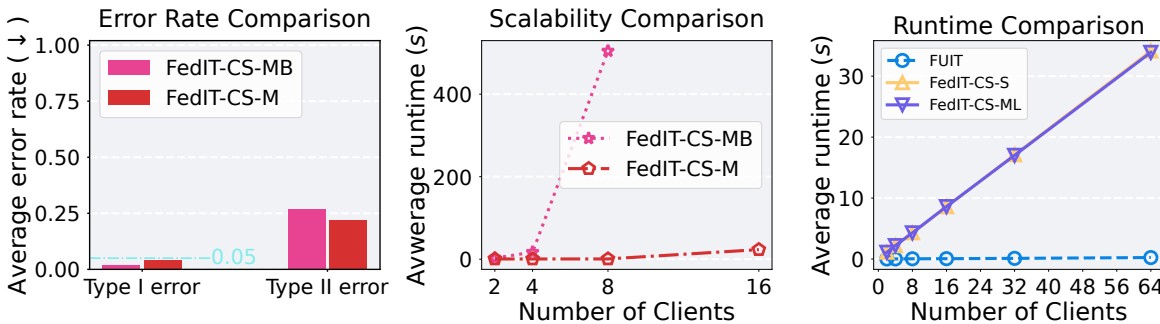

*Figure 9.* Efficiency and performance comparisons. Left and middle: performance and runtime comparison between FedIT-CS-MB and FedIT-CS-M. Right: average runtime comparison of the three linear-complexity methods in larger client settings.

## I.4. Runtime Evaluation

We further evaluate the runtime efficiency of methods whose aggregation cost scales linearly with the number of clients, namely FUIT, FedIT-CS-S, and FedIT-CS-ML. Following the linear setting in Sec. I.2, we fix the per-client sample size at $n = 200$ and vary the number of clients as $K \in \{2, 4, 8, 16, 32, 64\}$.

**Performance and analysis.** The results are reported in the rightmost panel of Fig. 9. All three methods exhibit approximately linear runtime growth with respect to the number of clients, and remain applicable in larger federated settings such as $K = 64$. FUIT achieves relatively competitive runtime performance. FedIT-CS-S and FedIT-CS-ML require permutation-based testing to obtain theoretically valid $p$-values, introducing an additional computational factor of roughly $B = 100$. We adopt this design because it offers strong empirical performance and broad applicability to arbitrary input distributions. Developing non-permutation alternatives, such as reliable distributional approximation methods, is a promising direction for further improving efficiency, although such designs remain technically challenging.

*Remark.* Although permutation testing incurs additional cost, the intra-client computations across the $B$ permutation samples are mutually independent and can therefore be parallelized. This substantially reduces the practical wall-clock overhead and improves the scalability of our framework in distributed settings.

## I.5. Results with More Real-world Validation

To further examine the applicability of our framework in more challenging real-world federated settings, we conduct a preliminary evaluation on the fMRI hippocampus dataset (Li et al., 2024). This dataset records brain activity from six

regions—PRC, PHC, ERC, Sub, CA1, and CA3—over $84$ consecutive days. We construct a federated setting by treating each day as a distinct client, resulting in $84$ clients in total.

**Experimental setup.** Unlike standard $i.i.d.$ data, this dataset exhibits strong temporal autocorrelation, evidenced by an ACF Lag-1 correlation of approximately $0.95$ and an exponentially decaying pattern. Such temporal dependence makes Type I error control substantially more challenging. As a reference, under the evaluation protocol described below, FUIT exhibits a markedly inflated Type I error rate of approximately $0.88$, highlighting the difficulty of this setting. To better account for the temporal structure in our permutation-based procedure, we adopt a circular-shift scheme, where each time series is randomly shifted as a whole along the temporal axis. This preserves the autocorrelation structure within each variable while disrupting cross-variable alignment, thereby producing more appropriate null samples for temporally correlated data.

For evaluation, we use domain knowledge (Bird & Burgess, 2008) to define benchmark pairs: (PHC, PRC) is treated as conditionally independent for Type I error assessment, while (CA1, Sub) is treated as dependent for test-power evaluation.

*Table 4.* Performance on fMRI hippocampus dataset.

| Method | Type I Error $K = 10$ | Test Power $K = 5$ | Test Power $K = 10$ | Test Power $K = 20$ |
|---|---|---|---|---|
| FedIT-CS-S | 0.11 | 0.78 | 0.93 | 1.00 |
| FedIT-CS-ML | 0.10 | 0.66 | 0.89 | 0.99 |

**Performance and analysis.** The results are summarized in Table 4. In this challenging temporally dependent setting, FedIT-CS-S and FedIT-CS-ML achieve Type I error rates of $0.11$ and $0.10$, respectively, when $K = 10$, substantially lower than the markedly inflated rate of approximately $0.88$ observed for FUIT under the same evaluation protocol. For the dependent pair (CA1, Sub), test power increases steadily as the number of participating clients grows from $K = 5$ to $K = 20$. In particular, both variants reach nearly perfect power at $K = 20$. These preliminary results suggest that our framework remains effective in structured real-world federated data beyond standard $i.i.d.$ settings.

### I.6. Comparison with Additional Baselines

I.6.1. COMPARISON WITH DENSITY-RATIO BASED BASELINE

To provide a broader empirical comparison, we additionally consider a baseline based on density-ratio estimation.

Prior work in centralized settings (Yamada & Sugiyama, 2010; Sugiyama & Suzuki, 2011; Sugiyama et al., 2012) has developed independence tests through density-ratio estimation, such as least-squares independence testing. Although these methods were not originally designed for federated learning and official implementations are unavailable, we adapt their core idea to construct a federated baseline. Specifically, we use the Squared-Loss Mutual Information estimator (SMI) and aggregate local SMI statistics across clients; we refer to this adapted baseline as SMI.

We compare SMI with our framework in three synthetic settings: *Cov* ($n_1 = 100$), *Freq* ($n_1 = 900$), and *Func* ($n_1 = 400$). For each configuration, we conduct $100$ independent trials and report the average results. The results are summarized in Table 5.

*Table 5.* Test power comparison against SMI under different synthetic settings.

| Setting | SMI | FUIT | FedIT-CS-S | FedIT-CS-ML | FedIT-CS-ML-F |
|---|---|---|---|---|---|
| Cov ($n_1 = 100$) | 0.71 | 0.16 | 0.78 | 0.63 | 0.83 |
| Func ($n_1 = 400$) | 0.24 | 0.65 | 0.80 | 0.61 | 0.83 |
| Freq ($n_1 = 900$) | 0.08 | 0.34 | 0.84 | 0.81 | 0.86 |

**Results and analysis.** SMI performs competitively in the linear *Cov* setting, achieving test power $0.71$. However, its performance declines substantially in the nonlinear settings, especially under frequency-domain heterogeneity, where the power drops to $0.08$. These results suggest that directly aggregating density-ratio-based local statistics without explicitly addressing client heterogeneity may be insufficient for robust federated independence testing in complex non-$i.i.d.$ scenarios. In contrast, our methods maintain consistently stronger performance across these settings.

### I.6.2. COMPARISON WITH ADDITIONAL CENTRALIZED BASELINES

To further assess the statistical power of our framework, we extend the comparison to adapted versions of two representative *centralized* independence tests: HSIC-Agg (Schrab et al., 2022) and dCor (Székely et al., 2007). These methods are applied by directly pooling client samples, which is incompatible with federated privacy constraints and may incur substantial communication cost. We include them solely as performance references.

**Results under the Cov setting.** We first evaluate performance under the *Cov* setting with varying local sample sizes $n_1$. As shown in Table 6, the centralized baselines are often less effective than our methods. This suggests that direct pooling under heterogeneous client distributions can weaken the aggregated dependence signal. In contrast, our FedIT variants, especially FedIT-CS-ML-F, exhibit substantially stronger sensitivity and achieve near-perfect power once $n_1 \geq 200$.

**Results in higher-dimensional settings.** We further consider a 4-dimensional scenario following (Gretton et al., 2007), where client-specific rotations $\theta \in \{0.2\pi/4, 0.3\pi/4, \ldots, 0.4\pi/4\}$ induce heterogeneous dependence strengths. The results in Table 7 reveal a clear performance gap. As the problem becomes higher-dimensional, the baselines become less effective; for example, at $n_1 = 600$, dCor and FUIT achieve powers of only $0.11$ and $0.09$, respectively. By contrast, FedIT-CS-S and FedIT-CS-ML-F maintain strong performance, reaching powers of $0.86$ and $0.83$. These results further support the robustness of our framework in heterogeneous and higher-dimensional environments.

*Table 6.* Test power comparison against adapted centralized baselines (HSIC-Agg, dCor) under the Cov setting.

| $n_1$ | FUIT | HSIC-Agg | dCor | FedIT-CS-S | FedIT-CS-M | FedIT-CS-ML | FedIT-CS-M-F | FedIT-CS-ML-F |
|---|---|---|---|---|---|---|---|---|
| 100 | 0.14 | 0.12 | 0.02 | 0.77 | 0.66 | 0.63 | 0.79 | 0.83 |
| 150 | 0.14 | 0.55 | 0.07 | 0.99 | 0.78 | 0.89 | 0.94 | 0.99 |
| 200 | 0.19 | 0.73 | 0.11 | 1.00 | 0.95 | 0.98 | 0.98 | 1.00 |
| 250 | 0.11 | 0.91 | 0.13 | 1.00 | 0.97 | 1.00 | 1.00 | 1.00 |
| 300 | 0.24 | 0.98 | 0.13 | 1.00 | 1.00 | 1.00 | 1.00 | 1.00 |
| 400 | 0.29 | 1.00 | 0.10 | 1.00 | 1.00 | 1.00 | 1.00 | 1.00 |

*Table 7.* Test power comparison in a 4-dimensional setting with heterogeneous dependency strengths (client-specific rotations).

| $n_1$ | FUIT | HSIC-Agg | dCor | FedIT-CS-S | FedIT-CS-M | FedIT-CS-ML | FedIT-CS-M-F | FedIT-CS-ML-F |
|---|---|---|---|---|---|---|---|---|
| 300 | 0.08 | 0.17 | 0.07 | 0.35 | 0.19 | 0.35 | 0.31 | 0.45 |
| 450 | 0.06 | 0.45 | 0.05 | 0.64 | 0.42 | 0.62 | 0.57 | 0.73 |
| 600 | 0.09 | 0.72 | 0.11 | 0.86 | 0.63 | 0.72 | 0.62 | 0.83 |

## J. Further Discussion

Our main formulation assumes a shared independence status across clients, as commonly adopted in related federated causality studies. To examine the scope of this assumption, we briefly consider a weaker setting in which only a subset of clients is dependent, which can be viewed as testing whether any client exhibits dependence. In this case, the weighting mechanism of FedIT-CS-ML may act as a soft selector by emphasizing more informative clients.

We consider $K = 5$ clients, where only two are dependent, with correlation coefficients $\rho \in \{-0.5, 0.5, 0, 0, 0\}$.

*Table 8.* Test power under heterogeneous dependence status, where only two of five clients are dependent.

| $n_1$ | FUIT | FedIT-CS-S | FedIT-CS-ML | FedIT-CS-ML-F |
|---|---|---|---|---|
| 100 | 0.07 | 0.65 | 0.51 | 0.80 |
| 200 | 0.13 | 0.98 | 1.00 | 1.00 |
| 300 | 0.23 | 1.00 | 1.00 | 1.00 |
| 400 | 0.24 | 1.00 | 1.00 | 1.00 |

As shown in Table 8, our methods remain effective and consistently outperform FUIT, providing preliminary evidence that FedIT-CS can extend beyond the shared-status assumption. More broadly, removing sample splitting and developing adaptive bandwidth selection are promising directions for further improving the power and practical performance of FedIT-CS.

