# OpenReview forum: "Powerful and Theoretically Guaranteed Independence Testing on Heterogeneous Federated Clients"
_ICML.cc/2026/Conference — ICML 2026 regular_

### Official Review · Reviewer_zfuR · 2026-03-13

**Soundness:** 3
**Presentation:** 3
**Significance:** 3
**Originality:** 3
**Overall Recommendation:** 4
**Confidence:** 3

**Summary:**

This paper studies federated independence testing under heterogeneous clients and argues that naive cross-client aggregation can both create spurious dependence under the null and wash out true dependence under the alternative. The proposed method, FedIT-CS, combines an empirical copula transform, random-feature CCA inside each client, a second copula transform to align client outputs, and then either subset selection or continuous client weighting for aggregation, with permutation testing for calibration and an HE-based privacy layer.

**Compliance With Llm Reviewing Policy:**

Affirmed.

**Key Questions For Authors:**

Can you provide a rigorous theorem for the implemented FedIT-CS-M / FedIT-CS-ML algorithms, rather than an oracle result that assumes a theoretically optimal aggregation algorithm?

Why does the permutation-validity proof remain correct through the CCA step? Please give a formal permutation-equivariance or exchangeability argument at the sample level.

What is the precise threat model and key-management protocol for the HE component?

**Limitations:**

yes

**Strengths And Weaknesses:**

Overall, the paper feels more like a promising journal-style framework than a conference-ready result. Too much results are in the appendix.

Strengths:
The problem formulation is meaningful, and the paper’s diagnosis of why pooling fails under heterogeneity is one of its strongest parts. The second copula transform is also an elegant idea: under the intended null, each client’s final outputs are independent Uniform[0,1], so simple mixing no longer induces artificial dependence.

The paper is generally readable. The progression from the failure cases in Figure 1 to the pipeline in Figure 2 is easy to follow, and the motivation is strong.

Weakness:
Theorem 5.1 is too weak relative to the claim: it assumes an oracle “optimal” aggregation rule and proves only a population-level sign result

The theory does not establish finite-sample validity, power improvement, or convergence of the learned weights

The privacy claim is overstated: saying the HE protocol “fully preserves privacy” is too strong when aggregated moments are still revealed to the decrypting client.

---

> ### Author Rebuttal · Authors · 2026-03-29
>
> Thank you for your positive feedback.
>
> **W1&W2&Q1**: Please see the reponse to Reviewer 6yk4 (W2&Q2).
>
> \
>
> **Q2**:
>
> Thanks for this point. We initially omitted these formal details, as we considered the measurability of the CCA solver to be a standard technical prerequisite. Specifically, as long as the procedure for identifying canonical directions is a **deterministic measurable functional**, the exchangeability of the samples is preserved.
>
> To clarify the exchangeability of the local CCA outputs, we provide a formalization of the procedure below. For simplicity, let the input matrices for the $k$-th client be $X\_k \in \mathbb{R}^{n\_k \times h}$ and $Y\_k \in \mathbb{R}^{n\_k \times h}$ (which correspond to the random projected vectors in our paper).
>
> 1. Formal Definition
>
> Let $Z\_k = (X\_k, Y\_k)$ denote the local dataset. We define the local CCA operator $\Psi_k$ as a mapping from the empirical data space to the canonical parameter space:
> $\Psi\_k: \mathbb{R}^{n\_k \times h} \times \mathbb{R}^{n\_k \times h} \to \mathbb{R}^{h} \times \mathbb{R}^{h} \times [0,1]$, $(u\_k, v\_k,\rho\_k) = \Psi\_k(X\_k, Y\_k)$.
> The operator $\Psi\_k$ identifies canonical directions via the SVD of the cross-covariance matrix $K\_k =C\_{XX,k}^{-1/2} C\_{XY,k} C\_{YY,k}^{-1/2}$. The canonical correlation $\rho_k$ is the leading singular value $\sigma\_{\max}(K_k)$, while the weights $u_k$ and $v_k$ are derived from the corresponding singular vectors.
>
> 2. Determinism and Measurability
>
> To ensure $\Psi\_k$ is a single-valued, Borel measurable functional, we introduce two standard technical conditions:
>
> Distinct Singular Values: We assume the underlying distributions are absolutely continuous, which ensures that the empirical covariance matrices have distinct singular values *almost surely*.
> Canonical Orientation Rules: To resolve the inherent sign and scale ambiguity in SVD/eigen-decomposition, we impose fixed orientation rules (e.g., enforcing unit variance $\\|u\_k\\|_{C\_{XX}} = 1$ and requiring the first non-zero entry of $u\_k$ to be positive).
>
> Under these conditions, the mapping $\Psi\_k$ is unique, Borel measurable.  While different numerical solvers may implement these constraints differently, any fixed deterministic solver satisfies the measurability required for our theoretical framework.
>
> 3. Exchangeability
>
> For any permutation matrix $P \in S\_{n\_k}$, let $X\_{k, \pi} =PX\_k$. We have $C\_{X\_\pi X_\pi, k} = \frac{1}{n\_k} (PX\_k)^T (PX\_k) = C\_{XX, k}$. Under the null hypothesis, the joint distribution satisfies $(PX\_k, Y\_k) \stackrel{d}{=} (X\_k, Y\_k)$, thus $C\_{X\_\pi Y, k} \stackrel{d}{=}C\_{XY, k}$，implying that the input to the SVD solver maintains the same distributional identity.
>
> **Theorem.** *Under $\mathcal{H}\_0$, the sequences of projected canonical variables are exchangeable.*
>
> **Proof.** Since $\Psi\_k$ is a deterministic measurable functional, we invoke the Push-forward Measure property. The mapping preserves the distributional identity of its inputs:
> $\Psi\_k(PX\_k, Y\_k) \stackrel{d}{=} \Psi\_k(X\_k, Y\_k)$
> This ensures that while the re-optimized weights $(u\_{k, \pi},v\_{k, \pi})$ vary numerically across permutations, their joint law is identical to that of the observed weights $(u\_{k, obs}, v\_{k, obs})$.
>
> Consequently, the resulting set of projected variables $R\_{\pi} = \\{(u\_{k, \pi}^T x\_{\pi(i)}, v\_{k, \pi}^T y\_i)\\}\_{i=1}^{n\_k}$ constitutes an **exchangeable random sequence** relative to the observed $R_{o}= \\{(u\_{k, o}^T x\_i, v\_{k, o}^T y\_i)\\}\_{i=1}^{n\_k}$.
>
>
> **Remark:** In practical implementation, since the CCA projection is followed by a Copula transformation, the output statistic relies purely on empirical ranks. This makes the procedure strictly invariant to monotonic scaling and robust to symmetric sign-flips, relaxing the burden on the orientation rules in empirical settings.
>
> We will refine these formal details in the revisited version.
>
> \
>
> **W3&Q3**:
>
> In FL, aggregated information (e.g., the global model or gradients) is typically shared, and thus we consider sharing aggregated moments with the decrypting client to be privacy-preserving. We will refine this claim in the revised version.
>
> Our threat model follows the standard honest-but-curious assumption, where the server adheres to the training protocol but may attempt to infer private information from shared messages. In our framework, the exchanged statistics are one-dimensional and highly compressed, making it difficult for an adversary to reconstruct the original data. Moreover, we employ HE to ensure that only aggregated statistics are exposed.
>
> Specifically, the encryption keys are generated by a trusted authority, and the private key is held by the client (or the trusted authority). Under this design, neither the server nor the decrypting party can access the raw statistics of any individual client, as only aggregated statistics are revealed throughout the protocol.

---

> > ### Author Rebuttal · Reviewer_zfuR · 2026-04-04
> >
> > I'll maintain my score of weak accept. However the paper still feels more like a promising journal-style framework than a conference-ready result.

---

> > > ### Author Response · Authors · 2026-04-04
> > >
> > > Thanks a lot for your feedback. We sincerely appreciated your time and effort in reviewing our paper and providing constructive comments. Frankly, we aren’t quite sure what you meant by “more like a promising journal-style framework than a conference-ready result.” We guess what you meant is that our work is more like a framework without enough technical details and sufficient convincing results. But for us, that is not true. Actually, due to space limit, we have to move a lot technical and implementation details, theorem proofs and experimental results to the appendices. Please note that our main paper has only 12 pages (including references), while there are 19 pages of appendices. We humbly ask for your attention to the appendices of our manuscript. Thanks again for your review and feedback --- Authors

---

### Official Review · Reviewer_6yk4 · 2026-03-16

**Soundness:** 2
**Presentation:** 3
**Significance:** 3
**Originality:** 3
**Overall Recommendation:** 4
**Confidence:** 2

**Summary:**

The paper tackles independence testing over heterogeneous, privacy-constrained federated data and proposes a method that uses copula alignment to normalize away marginal differences and stacked aggregation to combine local dependence evidence more effectively. The authors demonstrate their theoretical guarantees and empirical gains.

**Compliance With Llm Reviewing Policy:**

Affirmed.

**Final Justification:**

The rebuttal addressed most of my concerns, I'd maintain my positive score.

**Key Questions For Authors:**

- For HE, who has the secret key? That is who can decrypt? If it is the server, does that mean that the server can also see the encrypted data from the individual clients? You cannot compute on ciphertexts generated from different (pk, sk) pairs. It might be better to make this clear in the main paper that the clients share the same sk, and the server needs another round of communication to decrypt.
- Is there a gap between the theoretical analysis and the experiment you run? For theory you assume theoretically optimal aggregation, large sample size / many random features, and effective optimization. Is this stronger than the concrete finite-sample procedure you run? A finite-sample guarantee or an ablation on split ratio/cross-fitting would strengthen the paper.
- Do you assume all the clients share the same independence status? Is this assumption valid? How robust is the method when that assumption is violated, like when only a subset of clients is truly dependent, and can the weighting scheme work in this case?

**Limitations:**

I don't see negative societal impact.

**Strengths And Weaknesses:**

### Strengths
- The presentation is good. It's easy to understand the motivation and the challenge of heterogeneous settings.
- The paper proves soundness of the aggregated statistic in an idealized regime and gives Type I error control via the permutation test.
- The experiments are fairly broad

### Weaknesses
- The HE part needs better illustration.
- There is a gap between theoretical bounds and empirical settings.

---

> ### Author Rebuttal · Authors · 2026-03-29
>
> Thanks a lot for your comments. We present our responses below:
>
> **Response to W1&Q1**:
>
> Sorry for the confusion. The private key is held by the client (or a trusted authority), ensuring that the server cannot access the raw statistics of any individual client. Consequently, an additional round of communication is required to obtain the final result, as described in Appendix D (line 871). We will further clarify these details in the final version.
>
> **Response to W2&Q2**:
>
> We appreciate the reviewer’s rigorous assessment regarding the gap between theoretical oracle results and implemented finite-sample procedures. We offer the following clarifications on the technical challenges and the practical reliability of our framework:
>
> **1. Technical Barriers in Optimization Theory**
>
> The reviewer correctly identifies a gap between the theoretically optimal aggregation assumed in the theorems and the practical implementation using the **Adam optimizer**. In **FedIT-CS-ML**, the optimization objective is inherently **non-convex**. Proving the convergence of adaptive gradient methods (like Adam) to a global optimum in non-convex landscapes remains an open and significant challenge in optimization theory [1, 2]. Due to these technical limitations, a closed-form finite-sample guarantee for the specific trajectories of $p^*$ is currently beyond reach. This necessitates the "oracle" assumption to establish a foundational theoretical upper bound.
>
> **2. Signal Enhancement vs. Global Convergence**
>
> While we cannot analytically characterize the convergence to a global optimum, the validity of our method does not strictly require an "optimal" solution. It only requires that the optimization process achieves **signal enhancement**, meaning the learned weights $\mathbf{p}^*$ produce an aggregated statistic that is more informative than any single client’s local statistic.
>
> - **Empirical Robustness:** In practice, even if Adam converges to a local optimum, the optimization serves to amplify the dependence signal.
> - **Reliability:** Empirical reliability is maintained as long as the optimization successfully enhances the signal. Specifically, our theoretical framework, which posits $\rho > \max(\rho_k) > 0$, remains valid provided the optimization captures a stronger global dependency than any individual client’s local statistics.
>
> We acknowledge that a rigorous finite-sample theorem for the implemented FedIT-CS-M/ML would significantly strengthen the theoretical backbone. However, this requires bridging the gap between non-convex optimization theory and federated hypothesis testing—a direction we are actively exploring. We hope that as the community makes progress in characterizing the convergence of adaptive optimizers, we can further refine these bounds to replace oracle assumptions with constructive finite-sample guarantees.
>
>
>
> **Response to Q3**:
>
> Thanks for this insightful question regarding the assumption of shared independence status. While this assumption is standard in current federated causal discovery and independence testing works [3, 4], our framework is inherently **robust to its relaxation**.
>
> - **Conceptual Validity:** The testing objective can be generalized to detect whether **any** client exhibits dependence. Theoretically, our guarantees still hold under this relaxation.
> - **Weighted Aggregation Mechanism:** Unlike simple averaging, which may dilute local signals, our optimization-based weighting scheme ($\mathbf{p}^*$) naturally amplifies strong dependencies. If only a subset of clients is dependent, the model automatically assigns larger weights to those clients while down-weighting the independent ones. This acts as a "soft" selection mechanism, revealing which clients contribute most to the detected relationship.
> - **Empirical Robustness:** To verify this, we added an experiment with 5 clients where only a subset (two clients) is dependent (correlations $\rho \in \\{-0.5, 0.5, 0, 0, 0\\}$).
>
> | **n1** | **FUIT** | **FedIT-CS-S** | **FedIT-CS-ML** | **FedIT-CS-ML-F** |
> | ------ | -------- | -------------- | --------------- | ----------------- |
> | 100    | 0.07     | 0.65           | 0.51            | **0.80**          |
> | 200    | 0.13     | 0.98           | **1.00**        | **1.00**          |
> | 300    | 0.23     | **1.00**       | **1.00**        | **1.00**          |
> | 400    | 0.24     | **1.00**       | **1.00**        | **1.00**          |
>
> The results confirm that our method consistently achieves high power even under status heterogeneity, outperforming the baseline (FUIT).
>
>
> [1] Reddi S J, Kale S, Kumar S. On the convergence of adam and beyond[J]. arXiv preprint arXiv:1904.09237, 2019.
>
> [2] Défossez A, Bottou L, Bach F, et al. A simple convergence proof of adam and adagrad[J]. arXiv preprint arXiv:2003.02395, 2020.
>
> [3] Li et al. "Federated causal discovery from heterogeneous data." arXiv:2402.13241, 2024.
>
> [4] Huang et al. "Causal discovery from heterogeneous/nonstationary data." JMLR, 2020.

---

> > ### Author Rebuttal · Reviewer_6yk4 · 2026-04-02
> >
> > Thanks for the rebuttal. Most of my concrens are addressed. And I will keep my score. Please include the new results and discussions in the revision.

---

> > > ### Author Response · Authors · 2026-04-04
> > >
> > > Thanks a lot for your feedback. We appreciated your time and effort in reviewing our manuscript and providing constructive comments and suggestions. --- Authors

---

### Official Review · Reviewer_QMVW · 2026-03-19

**Soundness:** 2
**Presentation:** 3
**Significance:** 3
**Originality:** 2
**Overall Recommendation:** 4
**Confidence:** 2

**Summary:**

This paper considers the independence testing problem in federated setting with heterogeneous data distribution. While previous methods often encounter problems in this challenging scenario due to naive aggregation pitfall and dependence aggregation dilution, this paper proposes FedIT-CS that include copula-based marginal alignment technique to standardize diverse client distributions and a stacking-based aggregation strategy using Random Fourier Features (RFF) and Canonical Correlation Analysis (CCA) to extract the strongest local dependence signals without they being canceled out by inter-client variation. On the theory side, this paper shows that the aggregated statistic has bounded type 1 error. Empirically, FedIT-CS outperform baselines like FUIT on protein-signaling dataset.

**Compliance With Llm Reviewing Policy:**

Affirmed.

**Final Justification:**

The authors addressed my concerns and therefore I increase the confidence of my evaluation.

**Key Questions For Authors:**

How does the number of clients and the number of permutations B influence the type 1 error bound in Theorem 5.2?

**Limitations:**

Yes

**Strengths And Weaknesses:**

Soundness: The theoretical results (Theorem 5.1) look sound. The empirical results across diverse synthetic scenarios (Covariance, Frequency, Functional) and a well-known real-world benchmark (Sachs dataset) are reasonable.

Presentation: The overall presentation is good with clear motivation.

Significance: This paper address an important problem of federated statistic testing where existing methods fail to address the data heterogeneity problem.

Originality: I am not an expert in statistic testing. The overall paper looks original to me.

---

> ### Author Rebuttal · Authors · 2026-03-29
>
> Thank you for your positive feedback.
>
> **Response: Influence of $K$ and $B$ on Theorem 5.2**
>
> The Type I error bound defined in Theorem 5.2 is analytically independent of both the number of clients ($K$) and the number of permutations ($B$). The theoretical guarantee $P(\text{p-value} \leq \alpha | H_0) \leq \alpha$ remains valid due to the following reasons:
>
> - **Role of $B$:** While $B$ dictates the precision of the estimated p-value, it does not compromise the validity of the Type I error control. The p-value calculation in Algorithm 1 (Line 13) employs the standard estimator:
>
>   $$\text{p-value} = \frac{1 + \sum\_{t=1}^B \mathbb{1}\\{\rho^{P,\sigma\_t}_{xy} \geq \rho\^P\_{xy}\\}}{1 + B}$$
>
>   Under $H_0$, the rank of the observed statistic $\rho^P_{xy}$ among its $B$ permutations follows a discrete uniform distribution. The "+1" adjustment ensures that the test is **exact** or conservative for any $B \in \mathbb{Z}^+$. Crucially, if $B$ is small such that $1/(B+1) > \alpha$, the test becomes strictly conservative (Type I error might be 0), but the $\alpha$ bound is never violated.
>
> - **Role of $K$:** The number of clients $K$ affects the construction of the aggregated statistic $\rho^P_{xy}$. In a federated setting, increasing $K$ effectively increases the global sample size, which may enhance the statistical power (reducing Type II error). However, under the null hypothesis, the exchangeability of the samples remains intact regardless of $K$. Consequently, the probability of a false positive (Type I error) remains strictly controlled by the pre-defined significance level $\alpha$.

---

> > ### Author Rebuttal · Reviewer_QMVW · 2026-04-04
> >
> > My concerns have been adequately addressed

---

> > > ### Author Response · Authors · 2026-04-04
> > >
> > > Thank you very much for your feedback. We greatly appreciated your time and effort in reviewing our manuscript and providing constructive comments and helpful suggestions. --- Authors

---

### Official Review · Reviewer_rjhv · 2026-03-29

**Soundness:** 2
**Presentation:** 2
**Significance:** 2
**Originality:** 2
**Overall Recommendation:** 3
**Confidence:** 4

**Summary:**

This paper addresses federated independence testing (FedIT), which aims to determine independence relationships among variables across heterogeneous federated clients without sharing raw data. The authors identify limitations in existing methods (particularly FUIT), arguing that naive aggregation strategies ignore cross-client heterogeneity and lack theoretical guarantees under distribution shift.
The paper applies existing techniques — copula-based marginal alignment and stacking-based aggregation — to the federated independence testing setting.
The methods are evaluated on synthetic datasets and two real-world datasets (Boston housing, MNIST), showing power improvements over FUIT and adapted centralized baselines (HSIC-Agg, dCor) under heterogeneity settings.

**Compliance With Llm Reviewing Policy:**

Affirmed.

**Final Justification:**

The authors' rebuttal has partially addressed my concerns. In particular, the clarification regarding the copula transformation terminology and the additional explanations were helpful. However, I remain uncertain about the overall novelty and originality of the contribution. From my understanding, the proposed method primarily adapts existing techniques to the federated learning setting. While the paper is technically sound and the federated setting is practically relevant, I am not fully convinced that the level of innovation warrants a higher score. I have therefore raised my score from 2 to 3, reflecting that my concerns have been partially resolved but not entirely alleviated.

**Key Questions For Authors:**

### Q1:

In Section 3.1, Figure 1 describes two failure modes of naive aggregation: (i) variables that are independent within individual clients appear dependent after aggregation; (ii) variables that are dependent within individual clients appear independent after aggregation. The paper treats the **per-client (local) dependence relationship** as the ground truth, and considers any divergence between local and aggregated results as a "failure."

However, this perspective deserves further justification. If we pool all data together and view the combined dataset as a single global distribution, the dependence structure of this global distribution is also a valid notion of "truth." For instance, if client 1 has $\rho = 0.5$ and client 2 has $\rho = -0.3$, the pooled data may indeed appear uncorrelated — is this "spurious independence" or a genuine property of the global distribution? Similarly, if variables are independent within each client but the marginals differ across clients, the pooled data may show spurious correlation — but one could argue that this correlation is real from the perspective of the global mixture distribution.

**The core question is: what should the ground truth be?** Should we test the shared dependence relationship across clients (as Assumption 2.1 presupposes), or should we test the dependence in the global pooled distribution? The paper implicitly adopts the former, but does not adequately justify why. Please clarify and justify the problem formulation: what is the null hypothesis being tested, and why is the per-client consistency (Assumption 2.1) the appropriate starting point?

### Q2:

The paper applies copula-based marginal alignment as a core component of the proposed method. However, the Related Work (Appendix B) categorizes existing independence tests into three groups — rank-based, distance-based, and kernel-based — and does **not** mention copula-based independence testing, which is in fact a well-established line of research. The paper should clearly position itself within this existing body of work and explain what is genuinely new in the federated setting beyond the straightforward application of established copula-based testing ideas.


### Q3:

Equation (4) defines:

$$F(X) := F(X_1, \ldots, X_d) = C(F_1(X_1), \ldots, F_d(X_d))$$

Here the left-hand side $F(X)$ denotes the **joint CDF**, which maps $\mathbb{R}^d \to [0, 1]$ and returns a **scalar**. The copula $C$ also outputs a **scalar**. However, the subsequent text and the definition of the empirical copula transformation treat $F$ as a **vector-valued** function:

$$\hat{F}_n(\mathbf{x}) = \big[\hat{F}_{n,1}(x_1), \hat{F}_{n,2}(x_2), \ldots, \hat{F}_{n,d}(x_d)\big] \in [0,1]^d.$$

**Limitations:**

Yes.

**Strengths And Weaknesses:**

##  Strengths

**Heterogeneity consideration**: The paper explicitly addresses client heterogeneity in federated independence testing, which is a relevant and practical concern. The identification of how naive aggregation fails under distribution shift (Theorem 1) provides useful insight into the problem.

## Weaknesses

**Problem formulation is under-justified**: The paper implicitly treats per-client (local) dependence as the ground truth and considers divergence between local and aggregated results as a "failure" (Figure 1), but does not adequately justify why the shared local dependence relationship (Assumption 2.1) is the appropriate null hypothesis rather than the global mixture distribution. This is a fundamental conceptual issue that questions the paper's very premise — see Q1 for detailed discussion.

**Significant novelty deficit**: The core components — copula transformation for marginal alignment and stacking aggregation — are well-established techniques. Copula-based independence testing is an extensively studied line of research, yet the Related Work entirely omits this category and overstates the novelty of the copula component. The paper's contribution is thus primarily an adaptation of existing techniques to the federated setting rather than fundamentally new algorithms — see Q2 for detailed references.

---

> ### Author Rebuttal · Authors · 2026-03-30
>
> At the time approaching rebuttal close, we are unexpected to receive your review on our submission #25511 on **March 29, 2026**-**17** days after the review deadline (**March 12, 2026**) and **5** days after the rebuttal period began (**March 24**), and **only approximately 1 day** away from the rebuttal deadline (**March 30**). Anyway, thanks for your time and effort in reviewing our work. Following are our responses to your comments. Wish our feedback can address or clarify your concerns or possible misunderstandings.
>
> **About datasets mentioned in review Summary**
>
> Your comment: "The methods are evaluated on...two real-world datasets (**Boston housing, MNIST**)..."
>
> **Response:** This is NOT true. **Neither the "Boston housing" dataset nor the "MNIST" dataset was used or mentioned anywhere in our paper, experiments, or appendix.**
>
> \
>
> **Response to W1 & Q1:**
>
> This interpretation stems from a fundamental distinction between **statistical description** and **mechanism discovery**. While the pooled distribution provides a merely descriptive aggregation of the observed data, our work addresses Federated Independence Testing (FedIT) as a foundational building block for the causality literature. In this field, identifying invariant local independencies is a prerequisite for uncovering the underlying data-generating mechanisms.  We justify this formulation as follows:
>
> - **Consistency as the Ground Truth:** Our formulation aligns with the **Federated Causal Discovery (FCD)** paradigm (e.g., *Li et al., 2024*), where the "truth" is defined by the structural invariant rather than the observed mixture. In practical terms, FCD typically assumes that all clients share the same causal graph structure. Our assumption of consistent local dependence is a direct reflection of this standard setting.
> - **The Potential Pitfall of the Global Mixture:** While the global mixture provides a valid statistical summary of the pooled data, treating it as the "underlying truth" can be scientifically misleading in certain contexts. For instance, if a drug $X$ is independent of recovery $Y$ within every individual hospital (client), any correlation appearing in the pooled data due to **Simpson’s Paradox** would be regarded as spurious from the perspective of causal inference. Our method is designed to "see through" such heterogeneity to recover a consistent local truth, which is often a primary requirement for robust AI and invariant mechanism discovery.
>
> \
>
> **Response to W2 & Q2:**
>
> First, we cannot agree with you that our paper "**overstates the novelty of the copula component**".
>
> Characterizing our method as a 'straightforward application' of copula oversimplifies our contribution and ignores several key innovations unique to the federated setting:
>
> - **Novel Marginal Alignment for Heterogeneity:** We are the first to identify and resolve the failure of existing FedIT methods (e.g., FUIT) under heterogeneous settings. Our copula-based synchronization is specifically designed to handle distribution shifts—a challenge that does not exist in the centralized literature mentioned by the reviewer.
> - **Original Stacking-based Aggregation:** To the best of our knowledge, no prior work has proposed or analyzed a stacking-based aggregation scheme within the context of FedIT. This represents a distinct architectural contribution to the federated learning community.
> - **Theoretical & Empirical Rigor:** Beyond the application itself, we fill a critical theoretical gap by providing formal guarantees on how marginal alignment enables valid federated aggregation. This is further validated by a comprehensive empirical study.
>
> Second, it is NOT true that "the Related Work entirely omits this category".  We **did in fact cite** the relevant centralized copula work (e.g., Póczos et al., 2012, at Line 543). Our submission acknowledges these foundational methods while focusing on the distinct challenges of a federated setting, a differentiation that you neglected.
>
> \
>
> **Response to Q3:**
>
> This stems from a subtle distinction between two mathematically different objects. We have refined our notation to clearly separate the **scalar-valued** joint distribution from the **vector-valued** copula transformation:
>
> - **Joint CDF (Scalar):** $F(\mathbf{x})$ denotes the joint cumulative distribution function, mapping $\mathbb{R}^d \to [0, 1]$. By Sklar's Theorem, $F(x_1, \dots, x_d) = C(F_1(x_1), \dots, F_d(x_d))$.
>
> - **Copula Transformation (Vector):** We define the copula transform $\mathbf{F}(\mathbf{x})$ (and its empirical version $\mathbf{F}_n$) as a coordinate-wise mapping $\mathbb{R}^d \to [0, 1]^d$:
>
>   $$\mathbf{F}(\mathbf{x}) := [F\_1(x\_1), \dots, F\_d(x\_d)], \quad \mathbf{F}\_n(\mathbf{x}) := [F\_{n,1}(x\_1), \dots, F\_{n,d}(x\_d)]$$
>
> By using the **bold** symbol $\mathbf{F}$ for vector-valued mappings and standard $F$ for the joint CDF, the domain and codomain of each operator are now mathematically unambiguous.

---

> > ### Author Rebuttal · Reviewer_rjhv · 2026-04-05
> >
> > Thank you for the response. However, I remain unconvinced by the answers. Regarding Q3, the authors claim that $\mathbf{F}(\mathbf{x})$ is a "copula transform," but this is misleading — there is no copula function involved in the definition. Simply applying a coordinate-wise marginal CDF mapping $\mathbb{R}^d \to [0,1]^d$ does not constitute a copula transform; a genuine copula requires capturing the dependence structure via Sklar's theorem, which is absent here.

---

> > > ### Author Response · Authors · 2026-04-06
> > >
> > > Thank you for the response. We believe there may be a misunderstanding regarding the definition of a "copula transform."
> > >
> > > We understand the reviewer's concern that the definition of $\mathbf{F}(\mathbf{x})$ appears to be a coordinate-wise marginal mapping. However, in the context of copula theory and as established in [1], this specific mapping is formally referred to as the copula transformation [1] (see page 4 in [1]).
> > >
> > > The reviewer is correct that the operator $C$ does not appear explicitly in the expression $\mathbf{F}(\mathbf{x}) = [F_1(x_1), \dots, F_d(x_d)]$. However, the fundamental role of this transform is to isolate the dependence structure from the marginals. The resulting random vector $\mathbf{U} = \mathbf{F}(\mathbf{X})$ resides in the unit hypercube $[0,1]^d$, and its joint distribution is, by definition, the copula $C$. Thus, **the dependence structure is fully preserved and represented in the joint behavior of the transformed variables.** An intuitive visualization of this can be found in Figure 2 of [1].
> > >
> > > To ensure there is no further ambiguity, we have aligned our notation with [1] and provide the detailed definitions below:
> > >
> > > #### 1. Sklar’s Theorem and the Copula
> > >
> > > Let $\mathbf{X} = (X_1, \dots, X_d) \in \mathbb{R}^d$ be a random vector with marginal cumulative distribution functions (CDFs) $F_j(x_j) = \mathbb{P}(X_j \leq x_j)$. Sklar’s theorem states that the multivariate joint distribution function $F(\mathbf{x}) = \mathbb{P}(X_1 \leq x_1, \dots, X_d \leq x_d)$ can be represented as:
> > >
> > > $$F(\mathbf{x}) = C(F_1(x_1), \dots, F_d(x_d))$$
> > >
> > > where $C: [0, 1]^d \to [0, 1]$ is the $d$-dimensional **copula**.
> > >
> > > #### 2. Copula Transformation
> > >
> > > The copula $C$ is the joint distribution of the random vector obtained via the **copula transformation**. Let the mapping $\mathbf{F}: \mathbb{R}^d \to [0, 1]^d$ be defined as:
> > >
> > > $$\mathbf{F}(\mathbf{x}) := [F_1(x_1), \dots, F_d(x_d)]$$
> > >
> > > When applied to the random vector $\mathbf{X}$, the transformed variables $U_j = F_j(X_j)$ are uniformly distributed on $[0, 1]$, and their joint distribution is exactly the copula $C$. Hence, this mapping is the standard procedure to transform data into the "copula space."
> > >
> > > #### 3. Empirical Copula Transformation
> > >
> > > In practice, when marginal CDFs $F_j$ are unknown, they are estimated using the empirical distribution functions $F\_{n,j}(x) = \frac{1}{n} \sum_{i=1}^n \mathbb{1}\\{X_{i,j} \leq x\\}$. The corresponding mapping:
> > >
> > > $$\mathbf{F}\_n(\mathbf{X}) := [F\_{n,1}(X\_1), \dots, F\_{n,d}(X\_d)]$$
> > >
> > > is widely referred to in the literature as the **empirical copula transformation**.
> > >
> > >
> > >
> > > We hope these clarifications address your concerns and resolve any misunderstanding.
> > >
> > >
> > >
> > > [1] Póczos B, Ghahramani Z, Schneider J. Copula-based kernel dependency measures[J]. arXiv preprint arXiv:1206.4682, 2012.

---

### Decision · Program_Chairs · 2026-04-30

**Decision:**

Accept (regular)

**Comment:**

I have weighed up the reviews, rebuttals and subsequent discussion, factoring in the reviewer confidences and points raised by the authors.  The reviewers identified a mixture of strengths and weaknesses in this submission.
Strengths
* A clear problem formulation and set of results with associated guarantees
* Well-presented and clear to read
* Compelling experimental evidence supporting the claims on real benchmark data sets

Weaknesses
* Some concerns about the completeness of the work, details of the threat model, and robustness of the privacy claims
* Some technical details omitted which will help the completeness of the paper (e.g., permutation validity proof, impact of varying parameters)
* Could clarify more relationship with work on this problem in the non-federated setting

The weaknesses I judge to be minor, and can be addressed in a revision of the paper.  As a result, I believe that the paper just meets the criteria for acceptance in ICML.